# Demographics shape public preferences for carbon dioxide removal and solar geoengineering interventions across 30 countries

Benjamin K. Sovacool [1,2,3] ✉, Darrick Evensen[4,5], Chad M. Baum [1], Livia Fritz [1] & Sean Low [1,6]

Climate intervention technologies such as carbon dioxide removal and solar geoengineering are becoming more actively considered as solutions to global warming. The demographic aspects of the public serve as a core determinant of social vulnerability and the ability for people to cope with, or fail to cope with, exposure to heat waves, air pollution, or disruptions in access to modern energy services. This study examines public preferences for 10 different climate interventions utilizing an original, large-scale, cross-country set of nationally representative surveys in 30 countries. It focuses intently on the demographic dimensions of gender, youth and age, poverty, and income as well as intersections and interactions between these categories. We find that support for the more engineered forms of carbon removal decreases with age. Gender has little effect overall. Those in poverty and the Global South are nearly universally more supportive of climate interventions of various types.

Radical and frequently contested climate intervention technologies such as carbon removal and solar geoengineering are attracting increasing attention from researchers, investors, and policymakers as the adverse impacts of climate change are increasingly evident[1–3].

Carbon removal technologies including soil carbon sequestration, afforestation and reforestation, direct air capture, and bioenergy with carbon capture and storage may be employed to remove greenhouse gases from the Earth's atmosphere. These options are assigned, to a varying extent, an expandingly critical role within the range of strategies and trajectories that aim to reduce global temperature change or meet the longer-term targets embedded in the Paris Agreement[4]. Solar geoengineering technologies such as stratospheric aerosol injection, aimed at reflecting a portion of incoming sunlight back into space before it reaches the Earth's surface, could serve as a measure to slow the risks of global warming, or create a stop-gap period of adjustment that gives countries time to adapt to the impacts of climate change[5,6]. Other options, such as marine cloud brightening or cirrus cloud thinning, are being assessed for their potential to remediate the risk of pending "tipping points" in the climatic system, and to diversify the portfolio of options we must use to arrest increases in temperature[7].

Some commentators promote these options collectively to meet the goal of keeping climate change impacts well below 2 degrees Celsius, an ambition deemed still possible (although difficult)[8]. Others argue that carbon removal options are necessary to reach net-zero emissions targets, tackle the problem of residual emissions, or account for gaps in implementation inherent within the United Nations Framework Convention on Climate Change process[9]. Still others point out that insufficient climate action during the previous decade means that transformational development pathways are now required to reduce greenhouse-gas emissions at a scale of four times the work (greater emissions reductions) but one-third the time (stabilizing the climate by 2030, if not sooner)[10].

And yet, the public remains substantially unfamiliar with these technologies, restricting its ability to participate in ongoing discussions about science, policy, and deployment[11–15]. Research on demographic attributes such as gender, age, and socio-economic status remains particularly important, but also rarely examined. For instance, demographic aspects such as gender or income can be strongly differentiating variables that contribute to social vulnerability and that can help explain how the experiences of men, women, poor, and wealthy people differ during and after times of climate crisis, given that demographics shape cultural practices, social norms, work functions, and even access to security and resources of protection and safety. Women frequently confront conditions of vulnerability in multiple spheres (e.g., monetary poverty, hunger, unemployment, under-education) and are also more vulnerable to extreme weather events, to their impacts, therefore triggering situations of violence[16]. Current

¹Department of Business Development and Technology, Aarhus University, Aarhus, Denmark. ²Science Policy Research Unit (SPRU), University of Sussex Business School, Brighton, UK. ³Department of Earth and Environment, Boston University, Boston, USA. ⁴Institute for Global Sustainability, Boston University, Boston, USA. ⁵University of Edinburgh, Edinburgh, UK. ⁶Wageningen University and Research, Wageningen, Netherlands. ✉e-mail: benjaminso@hih.au.dk

research practices and technological designs concerning carbon removal and solar geoengineering tend to endorse masculine values of control and tend to produce gendered impacts which are only beginning to be understood[17–20]. Mahajan and colleagues "hypothesize that women will express greater concerns about solar geoengineering's unpredictability and that this concern will decrease support for its use and research when we control for other confounding variables"[21].

Youth are more vulnerable to the duration and severity of impacts of climate change than adults, given that they will generally live longer (incurring more exposure) but also presently have physiological factors (such as smaller lungs and less developed immune systems) that make particular impacts such as air pollution or heat stress more extreme. Tentative conclusions from the few studies that have examined youth perspectives on geoengineering have noted that younger people tend to prioritize climate action more strongly, but also to more strongly emphasize the need for international cooperation and governance[22,23]. It is also youth that are more likely to be on social media, a platform they can use to reach millions of other individuals when they discuss climate policy or technology[24].

Finally, concerns have been raised in the literature that carbon removal and solar geoengineering could constitute technological imperialism and colonialism[25–28], and could also have net positive (or negative) impacts on rates of global poverty[29,30]. Schneider even goes so far as to write that both forms of climate intervention are "bound to exacerbate concomitant socio-ecological and socio-economic global crises, deepen societal dependence on technocratic elites and large-scale technological systems and create new spaces for profit and power for new and old economic elites"[31]. Buck adds that "a critical reading views geoengineering as a class project that is designed to keep the climate system stable enough for existing production systems to continue operating"[32].

Many of the foregoing hypotheses and projections are theoretical expectations that have not yet been supported with robust empirical data. Drawing on a large-scale, cross-country set of nationally representative surveys (n = 30,284 participants, with at least 1000 in each country) in 30 countries and 19 languages, this article more rigorously and systematically examines public preferences for 10 climate-intervention technologies in relation to the demographic dimensions of gender, youth and age, and poverty and income. Because most of these technologies are novel, and several exist only at a conceptual level, the public have at present little understanding of them. Consequently, we needed to provide our survey respondents with factual descriptions of the technologies (see Supplementary Information). The responses to the technologies are built off of the foundation of these descriptions. These 10 technologies are:

- Stratospheric Aerosol Injection: this aims to limit the effects of climate change by using planes or balloons to spray small particles (aerosols) into the upper atmosphere;
- Marine Cloud Brightening: this aims to limit the effects of climate change by spraying small particles, such as sea salt, into the air over the oceans, to make clouds brighter;
- Space-based Geoengineering: this aims to limit the effects of climate change by putting a giant mirror or other reflective material in outer space between the Earth and the sun;
- Afforestation and Reforestation: both aim to limit the effects of climate change by planting trees;
- Soil Carbon Sequestration: this aims to limit the effects of climate change by changing agricultural techniques to store more carbon dioxide in soils;
- Marine Biomass and Blue Carbon: both aim to limit the effects of climate change by improving how much carbon dioxide is stored in the oceans;
- Direct Air Capture with Carbon Storage: this aims to limit the effects of climate change by using very large fans to remove carbon dioxide from the air;
- Bioenergy with Carbon Capture and Storage: this aims to limit the effects of climate change by growing and harvesting plants as a source

of energy and then storing the emissions permanently in rocks or underground reservoirs;
- Enhanced Rock Weathering: this aims to limit the effects of climate change by increasing the ability of rocks to absorb carbon dioxide from the atmosphere;
- Biochar: this aims to limit the effects of climate change by heating organic material, such as tree branches and cornstalks, inside a container with no oxygen.

Although much previous work has tended to look at each of these technologies by itself, or in comparison with only 2-3 other interventions[33], we examine all ten together as an integrated portfolio because this is how they may be synergistically deployed together as part of a future climate policy package, and because both suites of carbon removal and solar geoengineering technologies are shaping climate governance and mirrors the policymaking dilemma of choosing options with limited resources and uncertainty[34].

Our primary contribution rests on our contention that demographic attributes such as gender, age, or income could strongly relate to the perceived risks of climate impacts or preferences for energy or climate policy. It is demographic aspects of people (including preexisting conditions or patterns of deprivation) that serve as the key determinants of social vulnerability and the ability for people to cope with, or fail to cope with, exposure to heat waves, air pollution, or disruptions in access to modern energy services[35]. Moreover, the impacts of climate change are becoming increasingly appreciated by researchers and policymakers alike for being a deeply social problem, one that therefore needs further inquiry revealing the social factors that may accelerate, or block, engagement on this critical issue. Lastly, a fundamental reason for identifying demographic predictors is because those are proxies for what information different groups tend to think of (and how they think about it) when they evaluate emerging technologies[36].

Social and behavioral science research from multiple disciplines, including philosophy, psychology, communication studies, political science, and sociology, has yielded precious insight into the ways that gender, age or income can fundamentally shape public engagement with climate change and can interact with partisan and other sociocultural factors (e.g., individualistic and hierarchical worldviews) to influence how people perceive climate risks[37]. The present study highlights the critical utility of additional research examining how public perceptions of diversity and economic inequality both between and within nations color collective perceptions about climate change and radical climate interventions. Understanding points of support, or opposition, across different individual perceptions can more broadly reveal patterns of incipient social acceptance or social license to operate[38], or patterns of anticipated opposition[39], both of which have high relevance for decisionmakers. Demographic groups for whom the issue of climate change may be less politically charged, or those more willing to make sacrifices or act on climate change, represent critical audiences for bridging partisan disagreements and building consensus on policy.

## Definitions, terms, and positionality

Given the sensitivity of the topic, some definitions and reflections on gender, youth, and poverty are warranted.

Most simply, by gender, we refer to whether one identifies as male, female, or other, but we do recognize that gender includes a statement about biological aspects (one's sex) but also social and cultural aspects (one's social identity)[40].

We adopt the convention of the United Nations Department of Economic and Social Affairs to define youth as consisting of those between the ages of 15 and 24 years of age, although in our particular survey instrument, we include respondents between the ages of 18 and 24 years old (given that our ethics approval was not granted for minors below the age of 18). Implicit in this definition of youth is that it represents not only a range of ages but also a developmental stage demarcated by growing capacity and a broadening of perspectives as well

as growth in personality and maturity associated with moving into adulthood; this definition also appreciates the diversity of beliefs, values, worldviews, and expectations held by youth[41].

Finally, by poverty, we refer to those whose income falls below a minimum threshold of resources, e.g. a poverty line. But we do so with an appreciation that such a monetary definition does not adequately capture other forms of poverty including those focused on capabilities (deprivation of one's abilities to achieve a life they have reason to value) or social inclusion (the exclusion of particular groups from participating wholly and meaningfully in the society in which they live)[42]. That said, a monetary focus is well attuned to capturing many of the channels by which households escape or fall into poverty, including those related to income, prices, assets, productivity, and opportunity[43].

A corpus of scientific and media literature—some of which the study will present below—depicts the disproportionate and severe impacts that climate change and energy infrastructure development have on women, youths, or those in poverty. This framing can be implicit and complicit in presenting such groups as lacking agency and competence or depicting them in need of help or rescuing from others, including depictions of women as weak (as always vulnerable victims to climate change) or virtuous (as holding superior values and norms about the environment)[44,45]. This study subscribes to neither narrative, and instead represents the complex viewpoints of people identifying as women, youth, or in poverty in their own frames of reference. In sum: we aim to incorporate the complexity and variety of views and perspectives concerning climate change and nature, which is also intended to better reveal heterogeneity in values and preferences.

## Gender, youth and poverty in climate vulnerability and protection

This section summarizes insights from three different bodies of evidence, drawing from the broader work on climate protection, climate intervention, and climate change mitigation, including climate preferences and behavior. The extant literature tends to identify disparities in climate change impacts, and disparities in access or burdens related to low-carbon technology adoption, by gender, age, and poverty (inclusive of income and class), all of which are relevant for geoengineering and its climate interventions.

### Importance of gender

Climate change has gendered dimensions across themes as diverse as the impacts of climate change, disparities in concern over climate action, differing values and norms, and disparities in the adoption of, or impacts to, low-carbon technologies, policies and practices.

Firstly, women are much more likely to suffer death or injury from severe climate change events, and they are far more vulnerable to malnourishment and poverty when climate change threatens food and water security[46,47]. During droughts, it is women that are the most likely to starve—intentionally or unintentionally—when food insecurity becomes severe[48]. Women are known to have poorer resistance to changing disease vectors and disease outbreaks compared to men, especially when combined with poorer access to medical care and health services[49]. Women are also more prone to the impacts of extreme heat, given that they differ from men in their physiological compensation to elevated temperatures, and that women dissipate less heat by sweating, have higher working metabolic rates, and have other biological vulnerabilities to heat[50]. These vulnerabilities to heat become even more pronounced when women are pregnant, and prolonged exposure to high temperatures are even associated with a greater risk of menarche, still birth, congenital birth defects, and preterm delivery—regardless of maternal ethnicity or age, with younger mothers having an even greater risk of negative outcomes[50–52]. Furthermore, women experience greater deposition of inhaled particles in their lungs from air pollution, are more sensitive to toxicological exposure, suffer from higher rates of anemia, are at greater risk of violence (including sexual violence) and suffer disproportionate mortality and decreased life expectancy during and after disasters[53,54].

More generally, women are disproportionately affected by water scarcity, and tend to be more gravely impacted by water mismanagement, yet they face greater barriers than men in participating in water governance bodies[55]. An investigation into recovery efforts following the 2010 Pakistani floods revealed that not only were a majority of the victims women, but also that women systematically "were either overlooked in the distribution of relief or were unable to reach places of relief distribution due to social norms that restricted their mobility"[18]. Climate change impacts have a greater negative impact on the mental health of women, too, given that it is generally women who have caregiving responsibilities and disproportionately carry the burden of cleaning, cooking, and ensuring family wellbeing during disasters or floods, leading to significant mental trauma and stress[54]. Evidence has even revealed that climate disasters have led to increased cases of depression and suicides among women in the Maldives due to climate change related displacement and destitution[48].

Secondly, and relatedly, differences in concern for climate change exist between men and women. In an extensive and authoritative review of the literature, Pearson and colleagues evidenced a consistent gender gap in environmental concern in that women typically express greater levels of concern than men and demonstrate heightened perceptions of risks across a broad range of environmental hazards[37]. That same review noted that women have a greater likelihood of believing that climate change is real and caused by humans, perceive a greater number of climate change risks, express more knowledge about it, are less likely than men to endorse denialist claims, and are less likely than men to express skepticism about climate change on social media. This holds true across various cultures as diverse as Australia, Canada, Italy, the United Kingdom, and United States[37]. A recent survey in the UK focusing specifically on the threat of climate tipping points also established greater levels of concern among women[56]. Another study from the solar geoengineering literature identifies women (in the UK) as more likely to place trust in climate science[57].

Thirdly, another body of research emphasizes gendered values or norms—suggesting that women hold more pro-environmental or pro-sustainability values that they can transmit or pass onto others, especially their children[58–61]. "Gender Socialization Theory" suggests that "females tend to be socialized toward a feminine identity stressing attachment, empathy, and care, and males tend to be socialized toward a masculine identity stressing detachment, control, and mastery in many countries around the world"[62]. According to this theory, women also have a greater proclivity to express compassion, to show an "ethics of care," to be more nurturing, and to be more concerned about the needs of others as well as the needs of the environment or biosphere[37]. Relatedly, women are also found to be more averse to tampering with nature[63], a factor which has proven influential for predicting support for climate-intervention technologies[64,65].

Lastly, evidence reveals gendered disparities in technology adoption, or preferences for low-carbon practices or policies. Due in part to more restrictive gender roles and also in part to being more prone to poverty, multiple studies have shown that women in the Global South are less likely to adopt low-carbon agricultural practices or efficiency improvements on farms, including precision agriculture, ploughing, drought resistant seeds, or advanced management techniques[48,66–68]. A comparative lack of literacy to men and lack of ownership of land preclude women from pursuing a multitude of climate adaptation practices in the Global South[69], where women are generally integrated poorly into new technology sectors[70]. Women are often excluded from household energy decision making, but more immediately and severely suffer the impacts of energy insecurity; women also tend to lack the skills needed to maintain and repair innovations such as new cookstoves or solar home systems[71].

The patriarchal nature of gender relations in many cultures demand that women subsume responsibility for the private sphere and the household in nurturing and caring roles, thereby limiting women's freedom to assume positions of power or participation in the labor market, and reinforcing gender inequality in patterns of mobility[72–74]. It is also men who report greater usage rates for electric vehicles, greater chances for EV ownership, and greater distances traveled by cars[75]. Women are also

disproportionately affected by the burdens of electronic waste that arise from many low-carbon innovations such as solar panels or electric vehicle batteries, especially those that affect fertility and morbidity[76].

Violence is another category of harm differentiated by gender. Women are more at risk to technology abuse and even domestic violence pertaining to the adoption of smart homes, smart meters, household energy control systems and digitalization of energy practices[77,78]. Natural gas extraction and shale gas production, considered a bridge to low-emissions economies by some, also perpetuates increased rates of prostitution, sexually transmitted diseases and stillbirths, and erode food security, all which disproportionately affect women[79]. This is because it is women who are more at the risk of being coerced into sexual trafficking, who bear the burden of pregnancy, and who usually are responsible for food preparation in the home. The implication is that women could be more vulnerable to the violent impacts of any natural disasters or technology deployment caused by geoengineering.

Governance and lack of procedural governance is a final gendered dimension. Energy and climate policymaking around the world has also been critiqued for not adequately engendering the participation of women[80–82]. As Pearse summarized in their review, "in climate mitigation and adaptation projects in the Global South, women have comparatively few opportunities to participate in and influence decision-making"[40].

## Importance of youth and age

The literature on youth and age is not as extensive as that on gender, but the extant evidence does tend to focus on two areas: disparities in climate change impacts, and in technology adoption and preferences, especially a willingness to protest and undertake direct action.

Youth are more susceptible to the impacts of climate change than adults across a range of physical and mental dimensions of health. The World Health Organization suggests that children will suffer more than 80% of the injuries, illnesses, and deaths attributable to climate change[83]. The greater vulnerability of youth to climate change impacts—notably fatalities and injuries during disasters, heat stress, exposure to environmental toxins, and increased exposure to diseases in warmer temperatures—can be explained in part by physiology. This includes their less mature physiological defense systems, the fact that they interact with their environment more directly, that they depend on adults or others more for care, and that they accumulate risks and threats over a longer period of time (since their lifetimes are largely to unfold in the future)[84]. In Africa, it is youth who constitute the largest demographic group, and the largest labor force dependent on the land, but this only exposes such youth to the impacts that climate change is having on water quality and availability[85]. Psychological and mental health impacts abound as well for youth, including posttraumatic stress disorders, depression, anxiety, learning problems, sleep problems, and difficulties in learning[86]. Youth already struggling with depression and anxiety are at an elevated risk of worsening symptoms in the face of climate impacts, and young people are extremely vulnerable to depression when faced with climate-induced parental injury[87]. Troublingly, increased levels of domestic violence against youth and children have been reported following climate-change related events such as hurricanes, and education is jeopardized whenever extreme weather events destroy schools, or limit the ability for families to send their children to school[84]. Resource depletion and degradation of the environment have even been linked to violent conflict between youth groups over the scarce use of resources in places such as sub-Saharan Africa[88].

Although youth are historically underrepresented in decision-making processes, especially those below the voting age, they still possess differentiated preferences for technology adoption and disparate trends in preferences for climate action. Youth in many parts of the globe are more connected digitally, and more likely to independently assess climate change science and other information about the environment via the internet[89]. In India, youth are far more likely than adults to state that climate change is occurring, and to express awareness of major international organizations working on climate change[90]. A person's age can influence low-carbon mobility patterns and preferences as well. Multiple studies have found that

the relationship between age and transport emissions takes on an inverse u-shape with multiple turning points: both the young and old travel less than households in the middle with children[91,92]. Electric vehicle interest is higher among youth and younger adults, and that cohort also expresses the most familiarity with electric mobility as well as the greatest importance attached to the environmental impacts of automobiles[93]. Youth express greater knowledge and awareness than adults on things like willingness to use renewable energy[94], or literacy over electric mobility brands, performance, range, and price[95,96]. Youth are more likely to view ecosystem services as important and more likely to view nature tourism as a deeper healing experience[97]. Youth are also leading campaigns and "green carnivals" to promote energy efficiency or community based climate science efforts[98,99] as well as expressing greater trust in science in general[57]. It is low-income youth who are driving the adoption of solar energy in Tanzania[100], and youth groups associated with Indigenous peoples that are aspiring for a future with renewable energy rather than fossil fuels across India and the United States[101]. Finally, numerous studies have argued that youth are far, far more likely to take direct climate action, to protest or strike, and to join social movements committed to addressing climate change, envisioning such activism as dutiful and disruptive[41,85,102–104].

## Importance of poverty and socioeconomic status

Our last demographic dimension is that of poverty and socioeconomic status. According to the most recent data from the World Bank, approximately 9.2% of the world, or 719 million people, live in extreme poverty, or what the World Bank calculates as less than $2.15 a day, which makes them unable to meet basic needs[105]. Using a different estimation technique, 1.2 billion people in 111 developing or Global South countries live in multidimensional poverty, accounting for 19% of the world's population, including 593 million children. However, poverty is a Global North problem as well, with the same data suggesting that more than 37 million people were living in poverty in the United States, of which 11.1 million were children[105]. Poverty is intimately connected with class (people's economic or social status) and income (people's money, property, or financial resources). Such stark levels of poverty intersect with climate change in four meaningful ways. It creates disparities in carbon emissions, impacts beliefs on climate change, generates differential climate impacts, and reflects disparities in technology adoption.

Poverty, income, and employment can have strong effects on carbon emissions or knowledge about climate change risks. Multivariate studies that include income and employment status tend to note that unemployment or lower income is negatively associated with carbon emissions regardless of location[106], especially for home energy services such as heating[107]. Full-time employment and rising income tends to increase consumption levels which can increase both primary emissions and secondary impacts such as traffic congestion[91,108], contributing to disproportionately high emissions of people with high socioeconomic status[109]. Other studies have noted that when demographic variables such as race, education, or politics are accounted for, income still has a unique positive effect on whether people believe that climate change is occurring, and are knowledgeable about climate actions[37].

Moreover, attempts at theorizing why income and class shape decision-making or behavior have hypothesized that differential vulnerability and sensitivity to effects of climate change exist among individuals of lower socioeconomic status compared to individuals of higher socioeconomic status. That is, wealthier individuals may have lower risk perceptions related to climate change because they have the economic means to address threats posed by climate change, whereas poorer people might feel a heightened sense of vulnerability to negative impacts of climate change because they lack the financial means to address such threats[37]. Socioeconomic status—including both income and educational attainment—also predicts stronger partisan divides on climate change beliefs and risk perceptions[37]. Ballew and colleagues found in very large sample of U.S. adults (N = 20,024) that across all beliefs, higher education and higher income are very strong determinants of the degree to which individuals

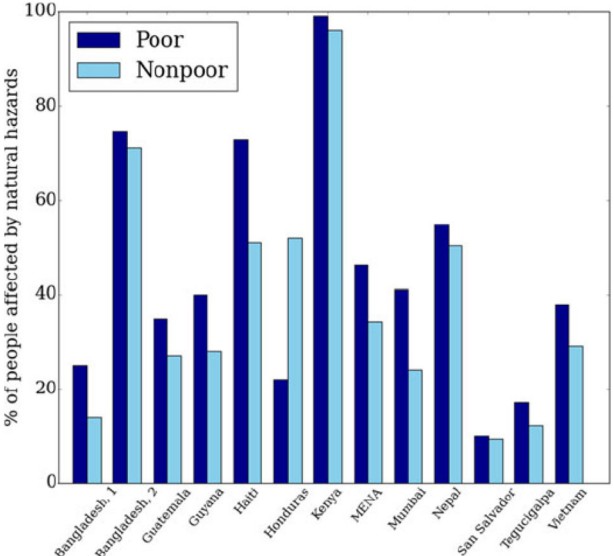

**Fig. 1 | Assessing the impact of climate hazards by groups self-identifying themselves as in poverty.** Source: Hallegatte, Stephane. Shock waves: managing the impacts of climate change on poverty. World Bank Publications, 2016. Note: Bangladesh 1 and 2 refer to separate studies done on the same country.

support climate policy or view climate change as a risk rather than an opportunity[110]. Bellamy also identified significantly greater concern about climate tipping points among higher "social grade" respondents in the UK[56].

Other research reports on how poverty and low socioeconomic status are key factors that increase the propensity for individuals and households to be physically harmed by climate change impacts, acting as a threat multiplier. As Leichenko and Silva write, "While climate change is never seen as a sole cause of poverty, research has identified numerous direct and indirect channels through which climatic variability and change may exacerbate poverty, particularly in less developed countries and regions"[42]. The reasons behind this heightened vulnerability are manifold, and include: lower income households have fewer assets to help them recover from climate shocks; depend more on climate sensitive sectors such as agriculture, forestry, fishing, or pastoralism for their livelihood; are more likely to live in areas of higher exposure to climate extremes; are less likely to have insurance; and are less likely to have the skills and capabilities to handle stress including higher levels of illness, mental stress, and stigmatization. As a case in point, poverty and income distribution are one of the most significant factors in determining one's vulnerability to food insecurity caused by climate change[111].

In addition, climate change can have longer, structural impacts on "poverty traps," the creation of self-reinforcing mechanisms such as market failures, inadequate legal protections, or even social norms that make it difficult for households to escape poverty[42]. Other studies have noted that "an increase in climate change vulnerability is positively associated with rising income inequality"[112] and that "poor people may be heavily affected by climate change even when impacts on the rest of the population remain limited"[113]. In Nigeria, the poorest 20% of the population are 50% more likely to be affected by a flood, 130% more likely to be affected by a drought, and 80% more likely to be affected by a heat wave than an average Nigerian[113]. The relationship between poverty and climate change can swing the other way as well. In India, a household affected by droughts in the past was 15 times more likely to fall into poverty[114]. Strong, consistent findings in the literature suggest that poor people are more exposed to environmental shocks and stressors and are more vulnerable to the impacts of natural disasters or hazards, losing generally a greater share of their assets than other socioeconomic groups. As Fig. 1 indicates based on a qualitative and descriptive study, whereas many studies have explored the exposure of poor

and non-poor households to climate hazards, all but one case found that the poor were more vulnerable than non-poor[115].

Additionally, poverty and socioeconomic status have been found in the literature to predict cooperation and pro-social behavior, and a willingness to adopt low-carbon solutions. For instance, a series of experiments concluded that compared to people from higher-social-class backgrounds, those from lower-social-class backgrounds—measured both in terms of resources and perceived class rank—were more charitable toward others[116]. Other work has noted that people's perceptions of their relative position in a social hierarchy, as well as subjective perceptions of resource scarcity and diminished rank, predict psychological motives, behaviors, and important life outcomes[117,118]. Still other work reveals vulnerabilities for low-income households who are unable to adopt new technologies, locking them into high-carbon and thus more vulnerable lifestyles. This encompasses those who are excluded from household solar energy schemes or electric vehicle charging due to lack of financial resources[119], or that risks from active transport—such as pedestrian and bicycle crashes and fatal cyclist crashes—tend to occur more often in low-income communities[120]. Research has also shown how more progressive or costly energy and climate policy, including carbon taxes, tends to disproportionately burden low-income homes[121].

## Research design
To investigate the prospective importance of gender, age and income on perceptions of climate interventions, this paper presents findings from a large-scale, cross-country set of surveys involving n = 30,284 participants in 30 countries (see Fig. 2). The surveys were nationally representative in terms of age, gender, and geographic region within those countries, and our approach also had quotas set for income and education. The survey instrument examined all ten climate intervention technologies, broken down into three technology groups: *SRM* (stratospheric aerosol injection, marine cloud brightening, space-based geoengineering); *ecosystem-based CDR* (afforestation and reforestation, soil carbon sequestration, marine biomass and blue carbon); *engineered CDR* (direct air capture with carbon storage (DACCS), bioenergy with carbon capture and storage (BECCS), enhanced weathering, biochar).

## Design of the survey instrument
Our survey instrument was conducted online across multiple platforms including those for mobile or handheld devices, as well as those using laptop or desktop computers. We ran the survey in a total of 30 countries with 19 languages (see Supplementary Table 1). Criteria for selecting countries included region, type of economy, population size, political organization, and carbon storage or solar geoengineering innovation potential, among others. Each survey had at least $N = 1000$ respondents for each country, and was nationally representative in terms of age, gender, and subnational geographic regions along with broad quotas for education and income.

The survey was designed to investigate public perceptions of climate-intervention technologies by means of different thematic dimensions such as perceived risks and benefits, support or lack of support for each climate intervention, support for various policy incentives as well as support for various policy restrictions, along with questions on sociodemographic characteristics, beliefs about climate change and environment, and trust in institutions and actors, and credibility of sources of information (Supplementary Table 2).

## Statistical analysis
Data were analyzed using SPSS v28.0. Descriptive statistical analysis included frequency distributions and comparison of group means. Significance testing employed Mann–Whitney $U$ tests (for gender, poverty or not, and Global North vs Global South). In all analyses, the dependent variables were support for climate interventions. One-way analyses of variance were used for assessing the relationship with age (these results were verified for robustness with non-parametric Kruskal Wallis H tests). Three-way (factorial) analyses of variance (ANOVAs) were used to test for

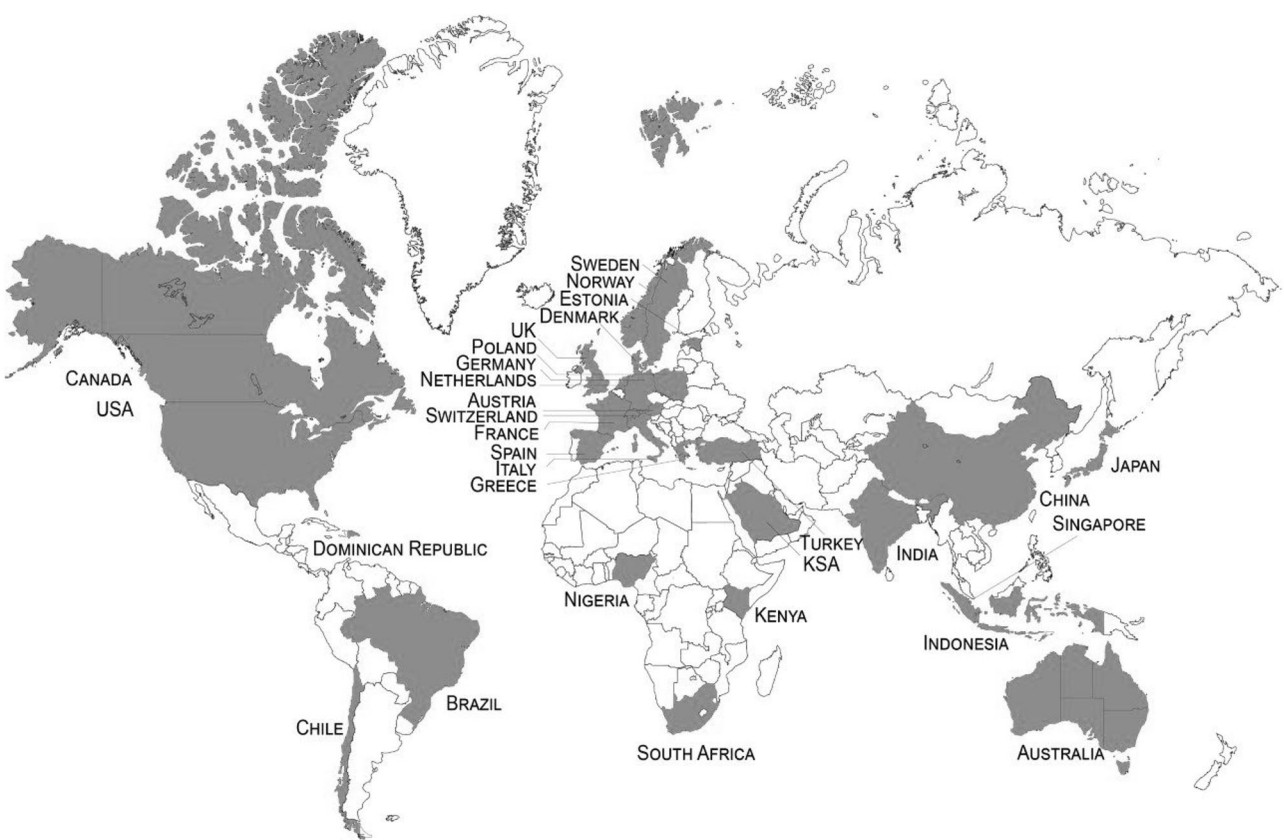

**Fig. 2 | Overview of 30 Countries Surveyed on Climate Intervention Technologies.** Note: Gray shaded areas indicate those where we conducted nationally representative surveys of the public, white color indicates no data. Diagram has been modified from Baum et al.[11].

interaction effects when including multiple independent variables in the same model. We ran ten three-way (factorial) ANOVAs with age, gender, and income (all binary) as the factor variables. We then ran ten additional three-way ANOVAs with age, gender, and Global North vs Global South (all binary) as the factor variables. Eta-squared (for ANOVAs) and r (for Mann–Whitney $U$ tests) effect sizes are reported throughout the analyses.

Given that the three support measures were strongly correlated (i.e., the lowest Spearman's rho correlation was 0.956 between small-scale field trials and broader deployment), we constructed a composite measure for support by taking the average of them. From a principal component analysis (varimax rotation), a one-factor solution was obtained for each of the ten technologies. Reliability was more than sufficient, with values of Cronbach's α for all technologies > 0.90.

### Ethical review statement

All components of the research were granted ethical approval by relevant authorities at Aarhus University. Full and informed consent was given by all participants before the beginning of the study, along with all participants being notified about the fact that their data would be handled in a fully anonymous manner and in complete accordance with the General Data Protection Regulation and any other pertinent data-security regulations, that any data would be analyzed in an aggregate fashion and would not be personally identifiable in any way, and that they had the right to withdraw their participation at any time. In addition, any questions about particular data being sensitive, including those that emerged in the course of the survey(s), were handled by erring on the side of caution and not asking a question in a given market. For instance, from the outset we decided not to ask about "political views" in China and the question on whether one self-identified as a "member of an ethnic minority or indigenous group" was removed in Estonia following feedback from participants.

### Contributions of the approach

By conducting surveys with such scale and scope, this exercise helps to provide a global baseline of SRM and CDR perceptions, in response to the information we provided about these approaches to SRM and CDR (see Supplementary Information). Given the newness and lack of public familiarity with the technologies, the determination was made to avoid 'priming' participants by valenced descriptions that overly focused on risks versus benefits, or vice versa, and as much as possible to talk about how technologies would work rather than what might go wrong, especially where significant uncertainty still prevailed[122].

The distinction between *ecosystem-based CDR* and *engineered or chemical CDR* might be imperfect, but we defend it on the basis that the categories entail different kinds of resource and energy demands as well as regarding how land is used, such that these differences may be significant across geographies and polities. We group carbon removal technologies based on the classifications and typologies in the literature offered by Morrow et al.[123], Low et al.[124], and Sovacool et al.[125] These all distinguish nature-based solutions (afforestation, soil management, blue carbon) from engineered solutions (biochar, enhanced weathering, DAC and BECCS). While there are obvious connections between the ecosystem-based and engineered carbon removal options, distinctions are made based on the degree of technical sophistication and maturity, capital intensity, and supply chains for carbon storage. Ecosystem-based approaches are those that feature a more prominent role of biological, ecosystem-based sinks with a relative focus on applications in terrestrial and marine environments. Engineered approaches differ by being more technological or chemical in nature, with a relatively stronger reliance on antecedent systems of resource extraction or mining, carbon capture and storage as well as transportation infrastructures. Biochar and enhanced weathering represent more hybrid approaches that blur these distinctions, but we classified them as more engineered than nature-based, at least in a comparative aspect. While we use

these categories for the presentation of our findings, we strictly avoided introducing any of the approaches to the survey participants as more or less natural, as engineered or ecosystem-based, thus attending to a potential framing effect or biasing related to "naturalness".

As participants were asked to evaluate multiple technologies within a technology category, this approach enables us to gain insights into relative preferences between technologies. Also, by asking members of the public for the first time in a survey on climate-intervention technologies about their support for different technological approaches, we can draw a distinction between the level and nature of support among the two types of CDR and SRM, as well as examine how such support varies across age, gender, and income. We must note that although each respondent only answered questions about either three or four technologies (i.e., one of the three technology categories), all respondents were randomly assigned to one technology category and approximately one-third of the respondents from each country were assigned to each category. Van den Brakel[126]. identifies randomized message treatments within probability samples as a means of simultaneously establishing strong internal and external validity (respectively via the random assignment and random selection). Furthermore, in 'a self-weighted sample design where sampling units are allocated proportionally to the treatments', Analysis of Variance tests (ANOVAs) can be used to examine differences across groups[126].

Our main dependent variable for all of our analyses is level of support for the ten climate interventions; for each intervention, we provide a composite value for support, generated from three measured support items (related to support for research, small-scale trial activities, and broad deployment, respectively). For extensive details about our research design, see Supplementary Information. Results of the full dataset, arising from the survey across the 30 countries, have been reported elsewhere (Baum et al.[11]); however, that prior analysis analyzed data only aggregated at the country level, whereas the focus in this article is heavily on the relationship between multiple individual-level demographic variables and support for the climate intervention technologies.

In terms of limitations, the survey instrument and its distribution across the 30 national samples offers extensive "horizontal" coverage without similar depth in terms of "vertical" coverage, either across time or having nested levels of geographical representation within nations. Although desirable in many contexts, a longitudinal design simply would not be possible with varying attrition rates across 30 countries and the necessary differences in how data collection needs to occur in different nations. In some contexts, it is not possible to reliably conduct repeat sampling, to say nothing of the expense. As we are seeking to identify intersectionally marginalized populations, including Global South nations is more important in sample selection than follow-up data collection with those members of the Global North about whom we already know the most from extant published research[33]. Another limitation

in data analysis is that each respondent only answered questions about one of the three technology groups. This complicates in some ways comparisons across the three groups, and introduces potential of methods effects affecting responses. Nevertheless, random assignment and ensuring that relatively equal number of respondents from each country were assigned to each technology group helps to mitigate such concerns.

## Results and discussion

For all our analyses, we considered ten dependent variables: these are composite measures of support for each of the climate intervention technologies. (See Supplementary Information for additional details on construction of these composite measures.) We provide data for the results for all dependent variables in the main text or the supplementary materials; only indicative results are present in figures in the main text due to the number of analyses run. We discuss all results in the main text. The means, standard deviations, and variance for the ten dependent variables are presented in Table 1. Mean support for three of the climate interventions was over 4.0 on the five-point scale, indicating individuals were somewhat supportive on average. For the other seven climate interventions, the mean lay between 3.0 and 4.0, indicating a level between 'neither reject nor support' and 'somewhat support'.

Again, note that for these relatively unknown technologies, respondent perceptions will be based heavily on the understanding they gained from the information provided in the survey (see Supplementary Information), and each respondent only answered questions in relation to one technology category—to keep the information provision and questioning to a reasonable length. Respondents were randomly assigned, in equal numbers from each country, to each technology category. Therefore, the respondents providing their views on SRM are different people from the respondents providing their views on the first set of CDR approaches, and again are different from the respondents assessing the second set of CDR approaches. The respondents rated each technology on a scale of 1-5 (strictly reject to strongly support); they were not asked to rank order their preferences for technologies. Whilst it remains possible that information from one technology could have influenced responses to the other technologies, or that questions about one technology led to an 'anchoring and adjustment heuristic', we believe the rating scale approach used always for reasonable comparison across all ten technological approaches.

We also tested for significant differences among the groups of respondents assigned to the respective technology categories (i.e., in terms of gender, age, education, income, living in an urban (versus suburban or rural) area, religiosity, political views, or self-identification as belonging to an ethnic minority or indigenous group). Having found no evidence for such significant differences, we thus identify no reason for any such extraneous biases on how the groups respond to the information provided.

**Table 1 | Descriptive statistics from our survey for support for ten climate intervention technologies**

| Climate intervention technology | Mean | Confidence interval, 95% | Standard deviation | Variance |
|---|---|---|---|---|
| Stratospheric Aerosol Injection ($n = 9943$) | 3.33 | 3.30–3.35 | 1.15 | 1.32 |
| Marine Cloud Brightening ($n = 9953$) | 3.50 | 3.48–3.52 | 1.12 | 1.24 |
| Space-based Geoengineering ($n = 9945$) | 3.40 | 3.38–3.43 | 1.20 | 1.44 |
| Afforestation and Reforestation ($n = 10002$) | 4.43 | 4.41–4.44 | 0.77 | 0.60 |
| Soil Carbon Sequestration ($n = 9973$) | 4.16 | 4.15–4.18 | 0.86 | 0.74 |
| Marine Biomass and Blue Carbon ($n = 9954$) | 4.13 | 4.11–4.14 | 0.87 | 0.76 |
| Direct Air Capture with Carbon Storage ($n = 9920$) | 3.73 | 3.71–3.75 | 1.04 | 1.07 |
| Bioenergy with Carbon Capture and Storage ($n = 9926$) | 3.73 | 3.71–3.75 | 0.99 | 0.98 |
| Enhanced Rock Weathering ($n = 9918$) | 3.48 | 3.46–3.50 | 1.11 | 1.24 |
| Biochar ($n = 9922$) | 3.85 | 3.83–3.87 | 0.97 | 0.94 |

Note: Support for the technologies was measured on a scale of 1-5: 1 = strictly reject, 2 = somewhat reject, 3 = neither reject nor support, 4 = somewhat support, 5 = fully support.

**Fig. 3 | Support for climate interventions by gender identified in our survey.** Note: Sample sizes for Fig. 3: SAI (female = 4981, male = 4924), MCB (female = 4990, male = 4925), Space (female = 4984, 4923), Afforest (female = 4954, male = 5012), Soil Carbon (female = 4937, male = 4999), Blue Carbon (female = 4931, male = 4986), DACCS (female = 4918, male = 4967), BECCS (female = 4920, male = 4971), ERW (female = 4917, male = 4966), Biochar (female = 4920, male = 4967). Note: Support for the technologies was measured on a scale of 1-5: 1=strictly reject, 2=somewhat reject, 3=neither reject nor support, 4=somewhat support, 5=fully support.

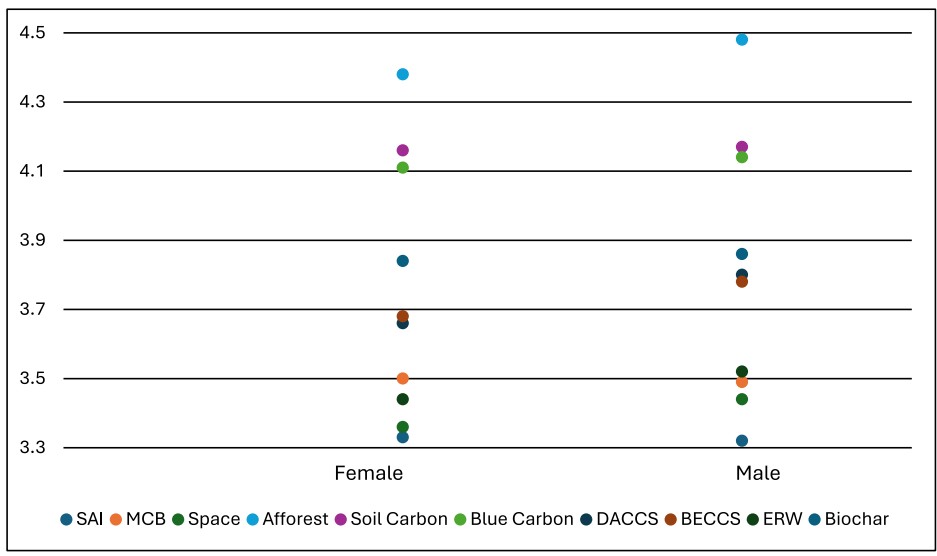

## Bivariate analyses: gender, age, and income

Our initial analyses examine the individual effects of age, gender, and income on support for the ten climate interventions. For parsimony of display, in all tables and figures, our ten climate interventions will be abbreviated as follows:

- Solar radiation management interventions: SAI (Stratospheric Aerosol Injection), MCB (Marine Cloud Brightening), Space (Space-based Geoengineering).
- Carbon dioxide removal group 1: Afforest (Afforestation and Reforestation), Soil Carbon (Soil Carbon Sequestration), Blue Carbon (Marine Biomass and Blue Carbon).
- Carbon dioxide removal group 2: DACCS (Direct Air Capture with Carbon Storage), BECCS (Bioenergy with Carbon Capture and Storage), ERW (Enhanced Rock Weathering), Biochar (Biochar Added to Soil).

**Gender**. We ran ten Mann–Whitney $U$ tests to examine variation in support for climate interventions between males and females (due to the quite small number of participants selecting "Other" or "Prefer not to say", these were excluded from the analysis). Due to our very large survey sample size, differences in support varied significantly (at $p < 0.05$, after Bonferroni corrections) between males and females for the following: Space, Afforest, Blue Carbon, DACCS, BECCS, and ERW. Nevertheless, as Fig. 3 indicates, in no instance was the mean difference in support larger than 0.14 (on a scale of 1–5) between the genders; effect sizes were uniformly small, ranging from an $r$ of 0.00 (for SAI) to 0.08 (for DACCS). In all six instances where the genders differed significantly on the level of support, males supported the intervention more than females.

There is often a presumption that men are more likely to prefer technical climate intervention than women, emerging from some earlier studies using smaller sample sizes in more limited national contexts. An older survey in the UK found that men tended to be more supportive of climate geoengineering[127], whereas a follow-up study by some of the same authors found no such difference[128]. Another survey in Switzerland in 2018 found that men were more likely to support DACCS and SAI[129]—though there was no effect for eight other technologies, including all ecosystem-based CDR options. In a more recent survey in the UK, men appraised engineered (DACCS, BECCS) and ecosystem-based CDR approaches (afforestation, wood in construction) more highly[130]. Men were also more likely to support a proposed DACCS project in a survey in the Pacific Northwest of North America—though only after receiving a tutorial on the need for carbon removal[131]. Meanwhile, a US survey conducted in 2019 identified women as most likely to support the use of soil carbon

sequestration with biochar—with no differences for the engineered CDR options (DACCS, BECCS)[64].

However, our findings contradict a nationally representative survey on solar geoengineering in the Fall of 2016 distributed to the United States electorate which found that support is higher among women than men[21]. Having inquired into specific attributes of the technology, Mahajan et al. posited that "women place more importance on the high speed and low cost of solar geoengineering than men do, whereas the importance of the risk of moral hazard and unpredictability does not differ across genders." Women in a 2013 German survey were similarly more supportive of solar geoengineering—though not afforestation or carbon capture and storage[132]. One of the very few prior cross-country surveys to include a Global South country (China), alongside five Western ones (UK, US, Canada, Germany, Switzerland) also found no variation in support by gender, in any country[133]. The same was also true for two enhanced weathering-specific surveys, one in the UK, US, and Australia, and one in only the UK[134,135]. In our sample, even when looking only at the United States sub-sample, males were still more supportive of every climate intervention, with the relationship statistically significant in two cases and the magnitude of difference higher in most cases compared to for the full international sample.

**Age**. All ten climate interventions show a clear effect of age in analysis of variance tests. Non-parametric Kruskal–Wallis $H$ tests revealed substantially similar results, based on significance, mean ranks, and eta² effect sizes. All three interventions in carbon dioxide removal group 1 (afforestation and reforestation, soil carbon, and blue carbon) reveal a negative effect of age on support, with youth (18-24 years) having significantly lower support than every other age category, after including Bonferroni corrections for multiple comparisons (Fig. 4). The opposite effect was true for the other seven interventions. For none of these seven interventions did any age group have significantly higher support compared to youth. For BECCS, ERW, and Biochar, youth support was higher than 55-74 year olds. For DACCS, Space, and MCB, youth support was higher than 45–74-year old. For SAI, youth support was higher than 35–74-year old. The effect of age on support was strongest for MCB and SAI (largest effect sizes, though all effects are "small" in nature; see Fig. 4).

In general, our findings support previous literature that shows a generally negative relationship between age and support for climate interventions[128–130]. However, the comprehensiveness of the current study, by including ten different climate interventions, adds nuance. We reveal a clear preference for more ecosystems-based interventions amongst older groups. Indeed, the actual level of support for the interventions shows that all age groups prefer the three ecosystem-based interventions the most (i.e.,

**Fig. 4 | Support for climate interventions by age category identified in our survey.** Note: Eta$^2$ values for ANOVAs on each intervention: SAI (.027), MCB (.032), Space (.022), Afforest (.011), Soil Carbon (.005), Blue Carbon (.005), DACCS (.021), BECCS (.013), ERW (.014), Biochar (.012). F statistics for all ten ANOVAs are significant at $p < 0.001$. Sample sizes for Fig. 4: 18–24 years = 4583, 25–34 years = 6569, 35–44 years = 6133, 45–54 years = 5627, 55–64 years = 4481, and 65–74 years = 2891. Each of the ten carbon removal technologies had between 9918 and 10,002 respondents. Note: Support for the technologies was measured on a scale of 1–5: 1 = strictly reject, 2 = somewhat reject, 3 = neither reject nor support, 4 = somewhat support, 5 = fully support.

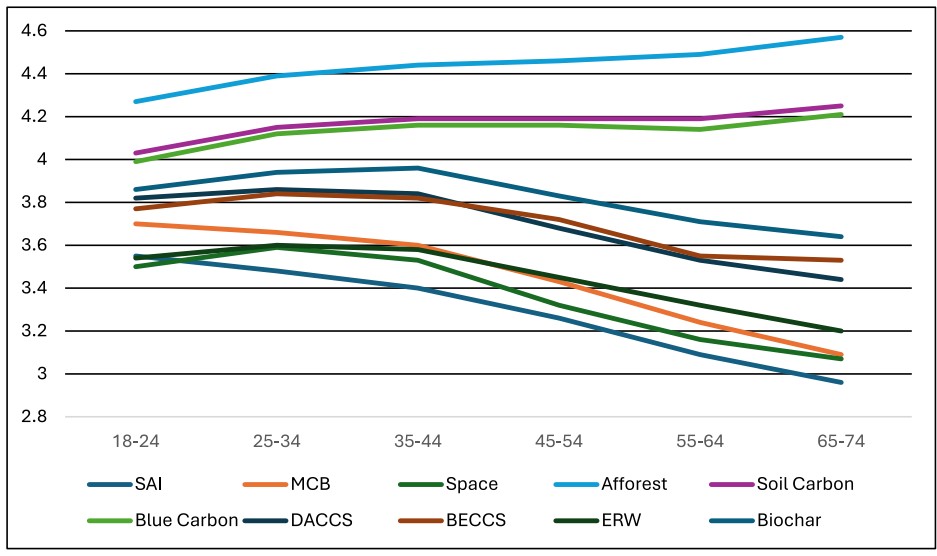

they have the highest mean in all six age categories). However, because support declines with age for the seven interventions and increases with age for the other three, this means the difference in levels of support according to age across the climate interventions grows increasingly large as people become older. This partially affirms earlier studies. In their own survey of the UK, Corner and colleagues found that older people tended to have less support for carbon removal and solar geoengineering[128]. They speculated that this could be because older participants were more unfamiliar with the options, or that they may have more experience with the hype cycles that often surround new technologies that don't end up being adopted. They also hypothesized that older participants are more likely to be skeptical of technological fixes.

At the same time, Carlisle and colleagues identified a similar pattern in a cross-country survey of Western countries (US, UK, Australia, New Zealand) of decreasing support by age for six engineered CDR and solar geoengineering approaches[136]. Similarly, a survey in the US by Sweet et al. found younger groups more generally supportive of CDR (except for afforestation)[64]. Adding to this, Spence and colleagues revealed in their cross-country survey that younger groups were more likely to support enhanced weathering in the US—there was no effect though in the UK or Australia[134]. Interestingly, in their survey on DACCS focusing on a potential project in the Pacific Northwest of North America, Satterfield and colleagues revealed that older age had a negative effect on support, but only before receiving a tutorial on the need for carbon removal – after this tutorial, the effect disappeared[131].

We do note that not all surveys are consistent (particularly for CDR); for instance, a survey in the UK found that older individuals tend to appraise engineered and ecosystem-based CDR options more highly[130]. However, looking narrowly at differences between age groups—since this is the only other study besides the present one (to our knowledge) to, e.g., distinguish those 18–24—the youngest group is never the one significantly appraising any of the options more negatively, but rather those slightly older (i.e., 25–34 or 35–44). In any case, Dunlop and Rushton found in their own work with young adults from Albania, Belgium, Czech Republic, the Netherlands, Poland, Portugal, and the United Kingdom that youth were more likely to have anxiety over climate change impacts, and to promote solutions to address it[23]. In their study, young adults talked about the importance of using geoengineering to "empower first, shame later" and that using geoengineering was like helping treat a terminal illness facing the planet. They lastly documented youth being frustrated with adults (the "older generation") for failing to take proper action on climate change.

**Income.** Ten Mann–Whitney $U$ tests examined variation in support for climate interventions between respondents in poverty and those not in poverty. We used a threshold of \$6.85/day, defined as the threshold for poverty in higher income countries by the World Bank (Table 2). Of the 30,284 respondents to our survey, 999 (or 3.3%) met this criterion. We acknowledge that what poverty means across the thirty different countries varies widely, and that there is likely notable additional variability within individual countries. We selected a relatively high poverty definition to capture as many relevant respondents as possible in this definition. Noting that all of our respondents identified as being in poverty come from only eight of the thirty countries, the country of residence clearly has an effect on whether someone meets our definition of being in poverty or not (see notes for Table 2). This poverty analysis is admittedly imperfect, but is a first attempt at offering empirical evidence that begins to shed light on the relationship between poverty and reactions to climate interventions—to help inform future research directions.

Differences in support varied significantly (at $p < 0.05$, after including Bonferroni corrections for multiple comparisons) between impoverished versus not for the following: SAI, MCB, Space, and DACCS (Fig. 5). These represent all three SRM methods and the most engineered CDR approach. Effect sizes were quite small for all four of the significant differences (ranging from $r = 0.03$ to $r = 0.06$). In all instances where those in poverty differed from those not in poverty, the respondents in poverty were more supportive of the climate interventions.

Surprisingly little research on perceptions of climate-intervention technologies has considered income, let alone poverty. Of the few studies that do, income tends to be included as a covariate, without the findings reported[137]. The above findings are thus something of a first in the literature, since the only research which to our knowledge considers class and income is restricted to the UK context. Of note, Bellamy found that appraisal of CDR options is higher among those of higher social grades, who also tend to report being more aware of climate tipping points[56,130]. Class, particularly in the rather unique setting of the UK, captures something quite different than income or poverty.

**Global South vs Global North.** Income, and whether someone falls below a poverty threshold, are individual indicators. Another more widely used societal-level indicator of economic wellbeing is whether someone is from a Global North or Global South nation. We split our sample into Global North ($N = 19,201$) and Global South ($N = 11,083$):
- Global North countries ($N = 19$): Australia, Austria, Canada, Denmark, Estonia, France, Germany, Greece, Italy, Japan, Netherlands,

**Table 2 | Poverty rates (% below $6.85/day) in study countries and in our sample identified by our survey[a]**

| | Country | % in country below $6.85/day | % in sample below $6.85/day | % in country below the national poverty line[b] |
|---|---|---|---|---|
| 1 | Nigeria (*n* = 1008) | 91 | 14 | 40 |
| 2 | Kenya (*n* = 1006) | 86 | 34 | 36 |
| 3 | India (*n* = 1018) | 84 | 0 | 22 |
| 4 | South Africa (*n* = 1016) | 62 | 9 | 56 |
| 5 | Indonesia (*n* = 1002) | 60 | 16 | 10 |
| 6 | Brazil (*n* = 1007) | 28 | 0 | No data |
| 7 | China (*n* = 1008) | 25 | 0 | 0 |
| 8 | Dominican Republic (*n* = 1002) | 23 | 15 | 21 |
| 9 | Turkey (*n* = 1024) | 13 | 4 | 15 |
| 10 | Chile (*n* = 1010) | 8 | 6 | 11 |
| 11 | Greece (*n* = 1005) | 4 | 0 | 20 |
| 12 | Spain (*n* = 1005) | 3 | 0 | 22 |
| 13 | Italy (*n* = 1002) | 2 | 0 | 20 |
| 14 | Australia (*n* = 1019) | 1 | 0 | No data |
| 15 | Austria (*n* = 1005) | 1 | 0 | 15 |
| 16 | Canada (*n* = 1005) | 1 | 0 | No data |
| 17 | Estonia (*n* = 1006) | 1 | 0 | 21 |
| 18 | Japan (*n* = 1011) | 1 | 0 | No data |
| 19 | Norway (*n* = 1002) | 1 | 0 | 13 |
| 20 | Poland (*n* = 1006) | 1 | 2 | 15 |
| 21 | Sweden (*n* = 1024) | 1 | 0 | 16 |
| 22 | United Kingdom (*n* = 1028) | 1 | 0 | 19 |
| 23 | United States (*n* = 1000) | 1 | 0 | No data |
| 24 | Denmark (*n* = 1010) | 0 | 0 | 12 |
| 25 | France (*n* = 1003) | 0 | 0 | 14 |
| 26 | Germany (*n* = 1025) | 0 | 0 | 16 |
| 27 | Netherlands (*n* = 1018) | 0 | 0 | 14 |
| 28 | Switzerland (*n* = 1003) | 0 | 0 | 16 |
| 29 | Saudi Arabia (*n* = 1002) | No data | 0 | No data |
| 30 | Singapore (*n* = 1004) | No data | 0 | No data |

[a]We calculated percentage of the sample below the $6.85/day threshold by using our survey data on respondents' monthly household income in local currency, converted to USD. This is a conservative estimate, because we only have data on household income, whereas the $6.85/day poverty metric relates to personal income. Additionally, for some countries (e.g., Brazil, China, and India), we report 0% in poverty because the lowest response category on our income variable has an upper limit too high to determine whether the respondent meets the poverty threshold or not. For example, the lowest income category for China is monthly income of less than 4000 yuan; however, 4000 yuan would equate to an income of 17.83 USD per day, which is substantially higher than the $6.85/day threshold. Consequently, we cannot determine whether any of our Chinese respondents fall below the poverty line which we designate. Therefore, only if the lowest category falls entirely below the threshold of $6.85/day, can we include respondents in that category as meeting the definition for poverty (e.g., Kenya's lowest income category in our survey is monthly income below 15,000 Kenyan shillings, which equates to a daily income of less than $3.71—placing the entirely income category below the $6.85/day threshold). However, if the upper limit of the lowest category extends higher than $6.85/day, we cannot reliably identify any survey respondents as impoverished.
[b]Data from World Bank (2023). Poverty headcount ratio at national poverty lines (% of population). https://data.worldbank.org/indicator/SI.POV.NAHC.

Norway, Poland, Spain, Sweden, Switzerland, Turkey, United Kingdom, United States.
- Global South countries: (*N* = 11): Brazil, Chile, China, Dominican Republic, India, Indonesia, Kenya, Nigeria, Saudi Arabia, Singapore, South Africa.

The results of the Global North vs Global South comparison, using independent samples t-tests, mirrored the findings from the income/poverty analysis, only the effect was much stronger for the societal-level tests (see Fig. 6). For Global North vs Global South, all of the differences in support for each of the ten climate interventions were significant (at $p < 0.001$, after Bonferroni corrections). The effect sizes were generally large, with Cohen's d values of MCB (0.57), Space (0.53), SAI (0.52), DACCS (0.47), BECCS (0.42), ERW (0.40), Biochar (0.34), Soil Carbon (0.24), Blue Carbon (0.19), and Afforest (0.06).

**Multivariate analyses**

To explore further the intersectionality amongst the core demographic variables in our analysis, we ran a series of three-way (factorial) ANOVA tests, with the climate interventions entered as dependent variables and three binary variables entered as factor variables: age (binary—youth or not), gender (male or female), and poverty status (yes or no). In each ANOVA, we examined whether the demographics, and interactions between each of the demographics, remained significant influences on support for the climate interventions, when accounting for the multiple joint influences.

Poverty was still a significant predictor of support for five interventions: SAI, MCB, Space, DACCS, and BECCS (see Figs. 7–9, and Supplementary Information). Youth was still significant for SAI, MCB, Afforest, Soil Carbon, and Blue Carbon. Gender was significant only for DACCS and BECCS. Across the ten climate-intervention options, three general patterns emerged.

**Fig. 5 | Support for climate interventions by income identified by our survey.** Note: Support for the technologies was measured on a scale of 1-5: 1=strictly reject, 2=somewhat reject, 3=neither reject nor support, 4=somewhat support, 5=fully support.

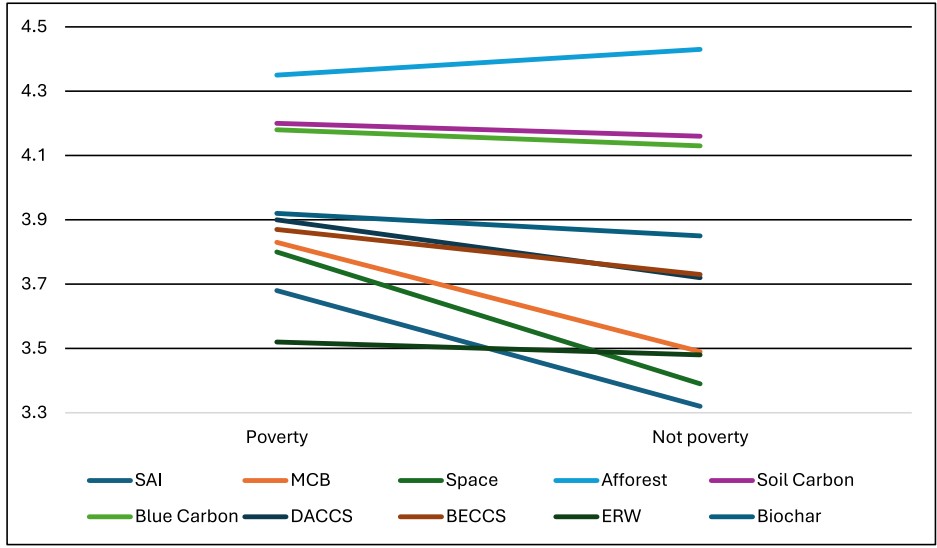

**Fig. 6 | Support for climate interventions by Global South vs Global North identified by our survey.** Note: Support for the technologies was measured on a scale of 1-5: 1=strictly reject, 2=somewhat reject, 3=neither reject nor support, 4=somewhat support, 5=fully support.

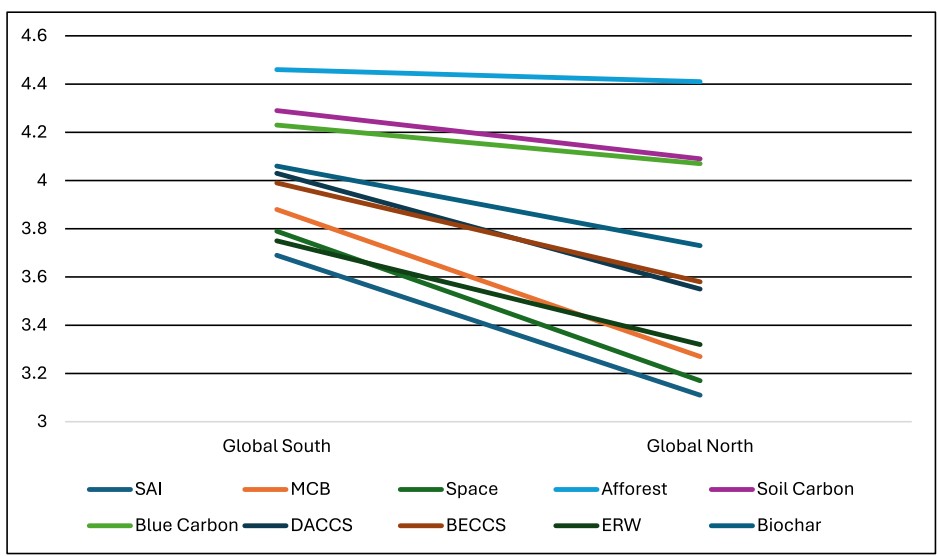

**Fig. 7 | Support for marine cloud brightening by age, gender, and poverty identified in our survey.** * In a three-way ANOVA, poverty (partial eta$^2$ = 0.02) and youth (partial eta$^2$ = 0.00) were significant (at $p < 0.05$), but gender, all three two-way interaction effects, and the three-way interaction were non-significant. F statistic for the ANOVA was significant at $p < 0.001$. Sample sizes: Youth not poverty female = 729, Youth not poverty male = 603, Youth poverty female = 68, Youth poverty male = 71, Older not poverty female = 4096, Older not poverty male = 4158, Older poverty female = 97, Older poverty male = 93. Note: Support for the technologies was measured on a scale of 1-5: 1=strictly reject, 2=somewhat reject, 3=neither reject nor support, 4=somewhat support, 5=fully support.

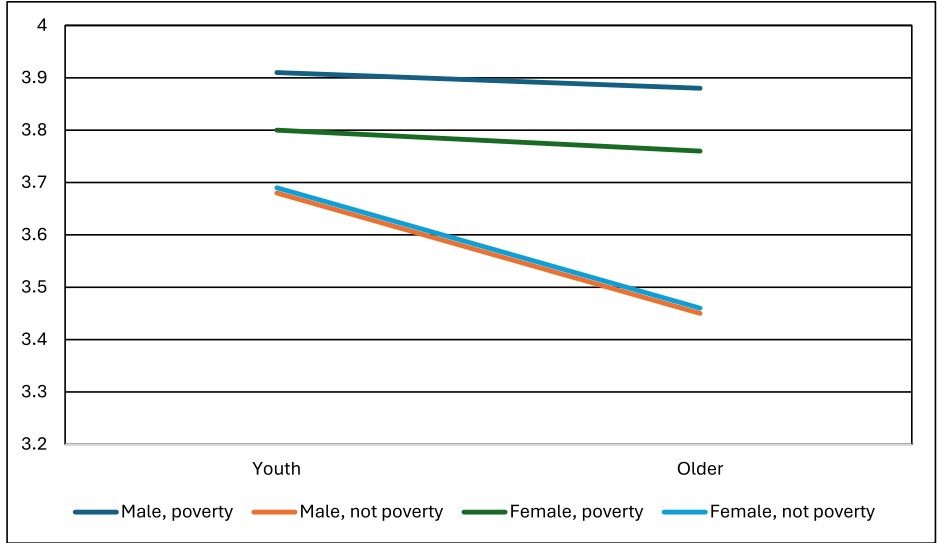

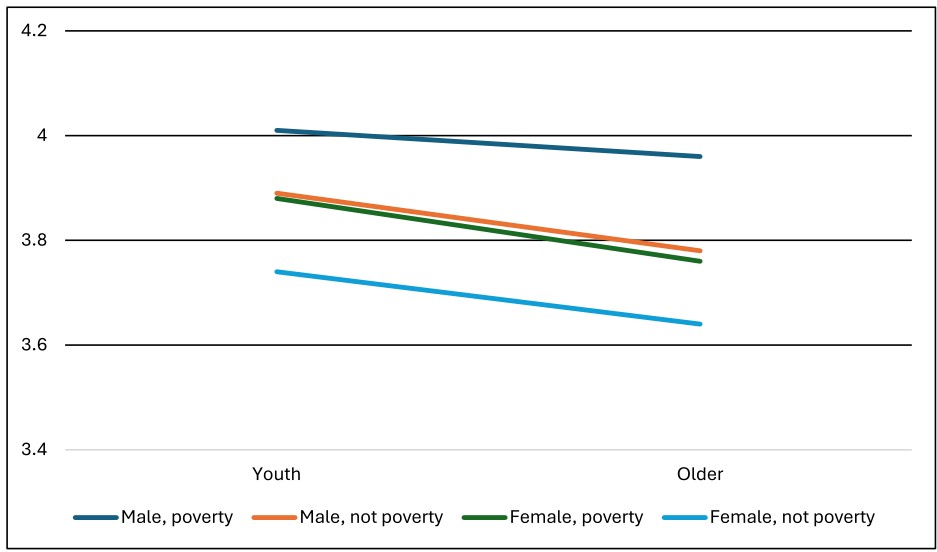

**Fig. 8 | Support for Direct Air Capture by age, gender, and poverty identified in our survey.** * In a three-way ANOVA, gender (partial $eta^2$ = 0.01) and poverty (partial $eta^2$ = 0.01) were significant (at $p < 0.05$), but youth, all three two-way interaction effects, and the three-way interaction were non-significant. F statistic for the ANOVA was significant at $p < 0.001$. Sample sizes: Youth not poverty female = 733, Youth not poverty male = 619, Youth poverty female = 75, Youth poverty male = 61, Older not poverty female = 4030, Older not poverty male = 4195, Older poverty female = 80, Older poverty male = 92. Note: Support for the technologies was measured on a scale of 1–5: 1 = strictly reject, 2 = somewhat reject, 3 = neither reject nor support, 4 = somewhat support, 5 = fully support.

First, there were instances in which poverty and youth both shaped support for climate intervention, but no interaction effects were present. This is seen in SAI and MCB (Fig. 7). Higher support comes from youth and those in poverty.

Second, there were instances in which gender and poverty both shaped support for climate intervention, but no interaction effects were present. This is seen in DACCS (Fig. 8) and BECCS. Higher support comes from men and those in poverty.

Third, there were instances in which interactions occurred, notably for the nature-based CDR category – afforestation, soil carbon, and blue carbon. For afforestation, support increases more for younger males than younger females (interaction between age and gender, Supplemental Materials). For soil carbon and blue carbon, the influence of being youth increases support more for those in poverty than not (interaction between poverty and age), see Fig. 9.

Beyond the individual-level effects of gender, youth, and poverty, we saw notable interaction effects when looking at Global South vs Global North. For SAI, support drops substantially as age increases in the Global North, but there is little difference across youth versus older cohorts in the Global South (see Fig. 10). A similar interaction effect is manifest for MCB, Space, DACCS, BECCS, ERW, and Biochar, with some of these even showing increases in support in the Global South increasing as age increases, whilst the opposite is revealed in the Global North (see Supplementary Information). A reverse interaction effect is also revealed for Soil Carbon and Blue Carbon – support in the Global North remains relatively stable across youth and older cohorts, but in the Global South, support for both of these interventions increases with age.

## Conclusions

Carbon removal and solar geoengineering options could become pertinent strategies for curtailing and even stabilizing greenhouse gas emissions, or lowering global temperatures by midcentury. We presented results from an original, first of its kind cross-country set of 30 nationally representative surveys (n = 30,284 participants, with at least 1000 in each country), with embedded random-assignment information conditions, to examine public knowledge and perceptions of these emerging climate intervention technologies. In doing so, we reveal complicated social dynamics behind how potential adopters and other members of the public hold views and preferences for nature-based climate interventions, engineered carbon removal options, and solar radiation management techniques. Our empirical results can inform ongoing discussions about energy and climate policy, the drivers of environmental change, and deliberations over future sustainability transitions.

Demographic attributes such as gender, age, and income feature crucially in explaining public preferences of climate-intervention technologies. On age alone, the standout observation is that for seven of the climate interventions support declines with age, and for three (the nature-based CDR options) support increases with age. Even in the three-way ANOVAs, used to explore interactions between the factors, the main effect of age is maintained in all three instances of older age relating to higher support. These three cases are clearly different from the others. The age differences matter for targeting communication about the approaches to CDR, and consideration of which policy options might be worth presenting to different audiences. It also suggests that over time, the seven CDR options where younger respondents supported the approach most heavily may come to see higher levels of support than they do today. These reactions, of course, depend on the information provided within the survey for these climate interventions. The information focused on functional descriptions of how the interventions work, concluding with a possible limitation on their potential effectiveness – in this way, we avoided making more valenced assessments of their merits and drawbacks. Nevertheless, as is necessarily the case when providing information, had different information been provided, respondent evaluations of the interventions could have differed. We have no reason, however, a priori or from the data itself, to suggest why the information we provided on the technologies would lead to any of these age-related effects.

We must note that different people answered the questions about the nature-based CDR options from the questions about the other climate-intervention technologies – due to each respondent only receiving information on three or four of the technologies. Nevertheless, due to random assignment to one of the three sets of technologies, and observing similar patterns amongst all seven other technologies even though they were in two separate groups, we believe the differences are robust that we see in how age influences support across the technologies.

Results from the Global North vs Global South three-way ANOVAs also importantly reveal that these age relationships for the seven engineered CDR and SRM interventions are stronger in the Global North, whilst the relationship for Blue Carbon and Soil Carbon is stronger in the Global South. The findings involving age are clearly relevant for decisionmakers developing communication strategies about climate change in general as well as those considering climate interventions. When interacting with youth in the Global North, a range of climate interventions can be targeted, but for older audiences in the Global North, there is a decidedly clear preference for nature-based solutions. In the Global South, support is higher overall and varies less across the age groups.

**Fig. 9 | Support for blue carbon and marine biomass interventions by age, gender, and poverty identified in our survey.** * In a three-way ANOVA, youth (partial eta$^2$ = 0.03) and the interaction between poverty and youth (partial eta$^2$ = 0.00) were significant (at $p < 0.05$), but gender, poverty, the remaining two two-way interaction effects, and the three-way interaction were non-significant. F statistic for the ANOVA was significant at $p < 0.001$. Sample sizes: Youth not poverty female = 706, Youth not poverty male = 661, Youth poverty female = 74, Youth poverty male = 77, Older not poverty female = 4049, Older not poverty male = 4152, Older poverty female = 102, Older poverty male = 96. Note: Support for the technologies was measured on a scale of 1–5: 1 = strictly reject, 2 = somewhat reject, 3 = neither reject nor support, 4 = somewhat support, 5 = fully support.

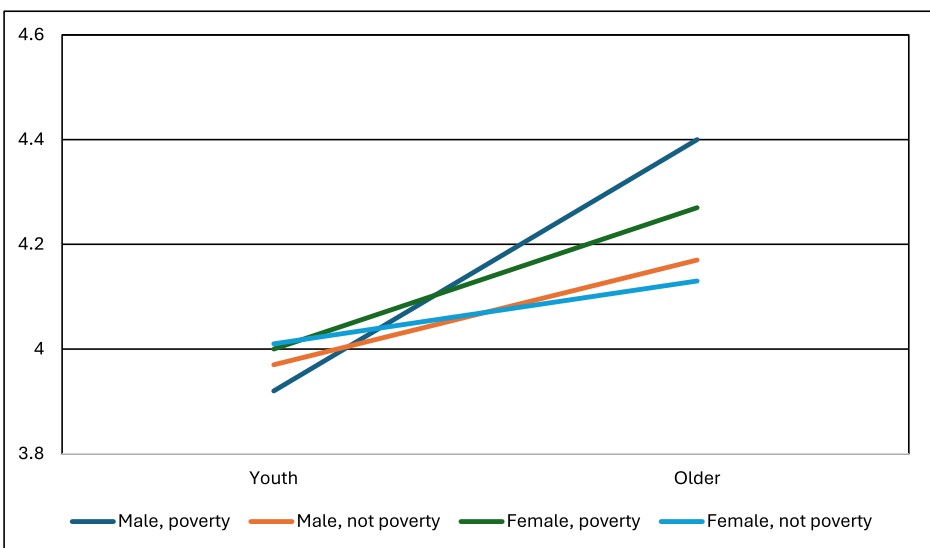

**Fig. 10 | Support for stratospheric aerosol injection by age, gender, and Global South vs North identified in our survey.** * In a three-way ANOVA, Global South (partial eta$^2$ = 0.021), youth (partial eta$^2$ = 0.003), and the interaction between Global South and youth (partial eta$^2$ = 0.002) were significant (at $p < 0.05$), but gender, the other two two-way interaction effects, and the three-way interaction were non-significant. F statistic for the ANOVA was significant at $p < 0.001$. Sample sizes: Youth Global North female = 426, Youth Global South female = 370, Youth Global North male = 329, Youth Global South male = 345, Older Global North female = 2770, Older Global South female = 1415, Older Global North male = 2728, Older Global South male = 1522. Note: Support for the technologies was measured on a scale of 1–5: 1 = strictly reject, 2 = somewhat reject, 3 = neither reject nor support, 4 = somewhat support, 5 = fully support.

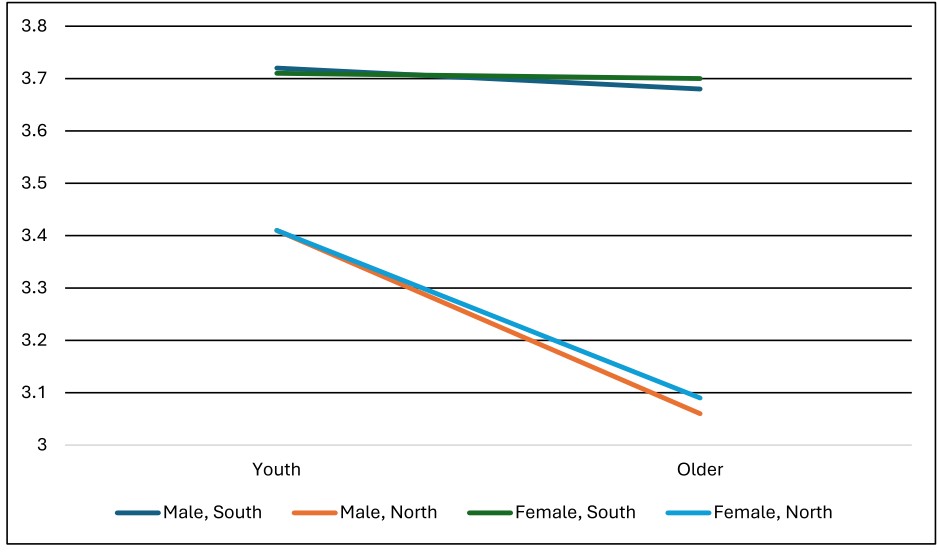

For gender, perhaps the most intriguing finding is how little effect this demographic characteristic had overall. All the effects from gender, when examined on its own, were small, and in the three-way ANOVAs the main effect of gender was relevant for only two of the ten climate interventions. Furthermore, a two-way interaction involving gender was only relevant for one climate intervention (afforestation). This finding is perhaps against expectations, given presumptions based on early survey research that men were more likely to support climate interventions, but it shows both the growing inconsistency of findings related to gender in the literature and the importance of robust empirical data to verify whether theoretical expectations are actually met or not. The lack of variation across gender is likely a beneficial finding when it comes to policy and communication, suggesting that these CDR approaches can benefit from support from both men and women, and that individual technologies are not seen as particularly problematic by either gender.

For income levels and poverty, the key finding is that on aggregate those in poverty were more supportive of climate interventions, compared to people not in poverty (observed for four of the ten interventions, with the other six interventions showing no relationship in either direction). This should be seen as a preliminary indication of a possible emergent relationship, due to the very coarse-grained manner in which we needed to define 'poverty' in our data set. Whether the respondent lived in the Global North or Global South shows the same relationship, only stronger. The Global South respondents supported each of the ten climate interventions more than the Global North respondents did. This seems to go against predictions based on literature about the effects on people in the Global South (e.g., reviewed earlier in this article). Especially for the three solar radiation management interventions, the effect of poverty, and of being from the Global South, on support are notable. Further research into the reasons behind the strongly positive effect of poverty and living in the Global South on support for geoengineering would be valuable, especially given the fundamental lack of such research to date. Subsequent research examining the effect of poverty on support for climate interventions should be specifically designed to over-sample from low-income populations and should

employ income thresholds in their demographic data gathering that align directly with individual national poverty levels, to allow for more precise comparisons between those in poverty and those not.

Nevertheless, even this initial, tentative finding has substantial policy relevance. Governments and government-industry partnerships seeking to deliver CDR projects in Global South and high-poverty areas can have more confidence than has been suggested by prior research that there is at least potential for public support for the CDR approaches. Of course, this does not in any way negate the necessity of fully considering and addressing concerns of procedural, distributive, and recognition justice, but it does dispel the a priori concern that CDR projects are simply more highly opposed in the Global South.

In relation to the more nuanced insights revealed in this study, it is notable that interaction effects (within the analyses using age, gender, and poverty) come only from the carbon removal interventions in group 2 (the nature-based options). The main take-home message here is that, whilst it is important to check for and understand intersectionality, it is the main effects of age, gender, and income that seem to be more important for support for climate interventions. For the nature-based interventions, we see some indication that the strong effects of age (here meaning that support is higher among the older cohort) are more pronounced in males and those in poverty. For the interactions effects in the analyses using age, gender, and Global South vs North, our repeated finding that age has different effects is valuable. It shows the importance of truly cross-national studies, and the inability to transfer simple lessons about demographic influences on climate change attitudes or beliefs between countries. It also points to the need for further research on why age-based dynamics operate quite differently across these macroscopic global regions.

## Reporting summary
Further information on research design is available in the Nature Portfolio Reporting Summary linked to this article.

## Data availability
The data that support the findings of this study are available at https://doi.org/10.5281/zenodo.13942571.

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

## Acknowledgements

This project has received funding from the European Union's Horizon 2020 research and innovation programme under the European Research Council (ERC) Grant Agreement No. 951542-GENIE-ERC-2020-SyG, "GeoEngineering and Negative Emissions pathways in Europe" (GENIE). The content of this deliverable does not reflect the official opinion of the European Union. Responsibility for the information and views expressed herein lies entirely with the author(s). Also, the project was approved by the Institutional Review Board at Aarhus University 2021-13 as well as by the Ethics Committee at the European Research Council.

## Author contributions

The study was conceived by B.K.S. The detailed design and materials were developed by C.M.B., L.F., S.L., and B.K.S., with C.M.B. and L.F. facilitating the data collection. C.M.B. and D.E. analyzed the data and B.K.S. wrote the paper with contributions from L.F., S.L., and D.E.

## Competing interests

The authors declare no competing interests.
