## [Peer Review file · Communications Earth & Environment]

Demographics shape public preferences for carbon removal and solar geoengineering interventions across 30 countries

Corresponding Author: Professor Benjamin Sovacool

Version 0:

Decision Letter:

Dear Professor Sovacool,

Your manuscript titled "The demographics of social preferences for carbon removal and solar geoengineering" has now been seen by 2 reviewers, whose comments are appended below. You will see that they find your work of some potential interest. However, they have raised quite substantial concerns that must be addressed. In light of these comments, we cannot accept the manuscript for publication, but would be interested in considering a revised version that fully addresses these serious concerns.

We hope you will find the reviewers' comments useful as you decide how to proceed. Should additional work allow you to

- address these criticisms (that is, either to incorporate the suggestions or provide a compelling argument why the point made by the reviewer is not valid or relevant to the editorial threshold as outlined below)

AND

- meet our editorial thresholds as outlined below,

then we would be happy to look at a revised manuscript.

In the following, we list our minimum requirements for publication.

*** Present novel and firmly supported insights into demographic dimensions of public preferences for carbon dioxide removal and solar geoengineering technologies.

*** Clarify the study scope and experimental design, including a selection of technologies and social variables, and provide details of your statistical analysis and significance.

*** Provide an in-depth discussion of your findings in the context of climate mitigation, communications, and policy and demonstrate that your claims are robustly supported (or you must tone down or remove claims that overreach what your data and the literature can support).

For publication in Communications Earth & Environment we request that you complete Nature Portfolio Reporting Summary (attached to this email). Please see our editorial policies page for details and to download the Reporting Summary document: <https://www.nature.com/commsenv/submit/submission-guidelines>

If the revision process takes significantly longer than three months, we will be happy to reconsider your paper at a later date, as long as nothing similar has been accepted for publication at Communications Earth & Environment or published elsewhere in the meantime.

Please use the following link to submit your revised manuscript, point-by-point response to the reviewers' comments with a list of your changes to the manuscript text (which should be in a separate document to any cover letter), a tracked-changes

version of the manuscript (as a PDF file) and any completed checklist:

Link Redacted

Please do not hesitate to contact us if you have any questions or would like to discuss the required revisions further. Thank you for the opportunity to review your work.

Best regards,

Martina Grecequet, PhD
Associate Editor,
Communications Earth & Environment
@CommsEarth

EDITORIAL POLICIES AND FORMAT

If you decide to resubmit your paper, please ensure that your manuscript complies with our editorial policies and complete and upload the checklist below as a Related Manuscript file type with the revised article:

Editorial Policy Policy requirements
(Download the link to your computer as a PDF.)

For your information, you can find some guidance regarding format requirements summarized on the following checklist: (<https://www.nature.com/documents/commsj-phys-style-formatting-checklist-article.pdf>) and formatting guide (<https://www.nature.com/documents/commsj-phys-style-formatting-guide-accept.pdf>).

REVIEWER COMMENTS:

Reviewer #1 (Remarks to the Author):

This is a tremendously ambitious effort to collect some of the most broadly representative data anyone has seen on perceptions of climate-engineering technology. This dataset is poised to become a focal point in all subsequent discussions of “social license” (or other preferred term). The initial report therefore has a tremendous responsibility, in my opinion, to be crystal clear about what conclusions it does versus does not support. It must therefore be held to the highest standards of rigor and transparency. While acknowledging the very high value of this effort at the outset, I elaborate on my key concerns along each of these dimensions below. My central concern is that so much hinges on one particular description of each technology and one particular judgment experience (within a cluster of other judgments). I think many readers will be inclined to think that there is some level of fundamental truth here because the samples are so big and so broad – a true strength! – but will fail to appreciate that these methods can say little about whether any technology judgment is robust to the exact description or order in which it came. This is not a fatal flaw but must, in my opinion, be discussed as a major limitation. My other major concern is that the reporting of methods, analyses, and results would be considered insufficient for behavioral-science outlets that I am familiar with. Before I elaborate on these concerns, I also discuss important questions about the scope and contribution of the current subset of analyses.

I would also contextualize my comments from the outset by saying that I do not work with such big datasets in my own work. I trust that colleagues with more statistical expertise will be consulted on the best approaches for modeling these results; especially in light of the choice to use a mixed design (with specific technologies grouped into technology categories).

I. Scope and Contribution.

I work mostly in a field where the prevailing norm is multiple studies per paper rather than multiple papers per study. Therefore, I do not know how Authors and Editors should negotiate the bounds of a single report based on such a large study. I will merely note that I think there are clear opportunities to learn substantially more about the stated research question(s) in this case. The authors have done an excellent job of pitching the importance of demographic factors in this societal discussion; but even in their own introduction to these issues, it is clear that the demographics are interesting insofar as they represent typical psychological concerns, needs, or access to information. Given the scope of the data-collection effort, it seems fair that a research team would reserve other questions for other reports. However, in my reading, we could potentially learn a lot more about why the demographic predictors show what they show by beginning to connect some of the psychological variables. For example, we might be able to learn a lot more about why age shows the patterns it shows by examining whether variables such as trust in science, sources of information, affect, or perceived risk account for the age finding. Similarly, we might learn a lot more about why poverty shows the patterns it shows by examining appropriate covariates as potential mediators. I acknowledge how subjective this suggestion might be – others who are more familiar with the norms here may feel quite differently – but it seems to me that the fundamental reason for identifying demographic predictors is because those are proxies for what information different groups tend to think of (and how they think about it)

when they evaluate these technologies.

II. Technologies versus Descriptions of Technologies.

I think it could set a very bad precedent to suggest in any form that this work allows comparisons or rankings of preferences between TECHNOLOGIES, per se. Research from many fields, disciplines, and traditions shows clearly that numerous details about the description can change people's perceptions (ranging, most relevantly, from risk information to specific word choices to comparisons with and contrasts to other technologies). The authors are careful not to offer direct statistical comparisons, nor interpretative statements that would advance this narrative; but they also state that the ability to compare across technologies is a potential strength and contribution of the work, and the figure design seems to invite these rankings. There are some very thoughtful discussions about this point in the Supplemental materials. I would suggest moving extended versions of these discussions to the main text to help readers avoid drawing overgeneralizations.

For example: "Furthermore, due to the novelty of the technologies, the determination was made to avoid 'priming' participants by descriptions that overly focused on risks versus benefits, or vice versa, and as much as possible to talk about how technologies would work rather than what might go wrong, especially where significant uncertainty still prevailed." It seems that this needs to go front and center in main text. There is no "neutral," so, of course, researchers have to make a choice. But it should be acknowledged that they are studying responses to [the chosen descriptions of the technologies] not responses to [the technologies themselves]. In other words, this single study, using one description per technology, is not able to do much to increase confidence that the absolute levels of acceptance would generalize beyond the specific stimulus used in the current study. It also seems critical to me that technology descriptions (or, at the very least, representative examples) should appear in the main text.

Another important point is provided later in the supplementals: "Probably the clearest example is 'stratospheric aerosol injection', given that this text is among the shortest and does not place substantial emphasis on possible impacts on the biosphere and weather patterns.... Still, we acknowledge that, by trying to provide a balanced presentation of risks and benefits and to provide participants broad information from which they could draw their own inferences, in this case, it may have unintentionally resulted in an insufficiently comprehensive illustration of potential risks." These caveats seem critical to address head on in the main text.

I would also suggest that categories of technologies be presented in separate figures or panels to further discourage direct comparisons given the countless subjective choices that limit direct comparison.

III. Experimental Design.

Treating the ten technologies as independent items (conceptually and statistically) does not seem appropriate given the experimental design. As explained in the Supplemental materials, the authors were sensitive to the possibility of contrast or assimilation effects – they refer to the general possibility as anchoring – but then, as far as I can tell, they treat tests associated with each technology as independent from one another.

It seems that most respondents are not very knowledgeable about the technologies. Therefore, they are likely to be looking for contextual cues. These cues might come from comparisons between technologies (As but one example: Marine cloud brightening might look less concerning /more favorable after thinking about SAI than after thinking about BECCS); and these cues might also come from generalizations across the category (for example: my intuition is that the analogous graphics for all three SRM technologies might suggest that they are basically the same but at higher altitudes). This calls into question the independence of the specific judgements; and potentially limits the external generalizability of any given result.

A conservative approach might be to treat the three or four items as multiple items from a scale, leaving three categories of responses. Do measures of reliability support this approach?
At the very least, order effects should be explored carefully.

With respect to these issues, there was some ambiguity about many specific experimental procedures. Within technology category, were specific technologies presented in a randomized or counterbalanced order? Or were they always presented in the same order? If they were balanced in some way, then robustness checks could test whether all the effects hold when looking only at the first technology that a participant rated. This would be extremely valuable for any discussion that purports to compare the technologies head-to-head. If the orders within category were fixed, then some technology-to-technology comparisons are inevitably confounded with potential order effects.

Do statistical analyses account for the technologies within groups? These aren't ten independent tests, they are three independent tests (SRM, Eco-CDR, Engineered-CDR). What are the alphas within categories? Would it make sense for the main analyses to be based on three categories rather than ten technologies?

IV. Operationalization of Poverty.

Were there any other measures of wealth or socioeconomic status used besides the very strict dichotomous World-Bank measure? It seems we could learn a lot more about the predictive power of wealth if additional analyses explore (a) continuous measures of wealth, and (b) relative measures of wealth. I am asking about (b) because subjective SES matters too. Income level X may make one relatively "rich" in one country and "poor" in another country; and it is possible that status offers unique predictive power beyond poverty. Perhaps a country-level percentile measure would be worth exploring.

V. Composite Measure.

I was surprised by the decision to collapse the three DV items, research / small-scale trial / broad deployment. I was even more surprised when I found the main result in the supplementals that apparently led to this decision: alpha values above 0.90. This seems like it might merit discussion in the main text, both because of the methodological importance and the potential context that it provides for interpretation. If participants are distinguishing between research and large-scale deployment at such low levels, this perhaps suggests that responses reflect something more like general attitudes than deliberative reflection.

Do people from some demographics distinguish between these three measures more than others do? If so, this would be an interesting finding in my opinion.

VI. Reporting and Transparency:

I am not familiar with this journal's reporting norms. As a true generalist journal, I imagine there is quite high variance. I was particularly surprised, however, that almost no descriptive statistics were reported in the main text or supplementals as far as I could tell. Readers cannot even find anything about variance, for example. This makes it difficult to have any intuition for the effect sizes. I was also surprised that there was very little detail on the inferential statistics aside from summary p-values. The amount of detail would be unacceptable in most behavioral-science outlets that I am familiar with. They may be appropriate here – I do not know.

- a. Was there pre-registration of analysis plans?
- b. How many participants were excluded?
- c. Are overall demographics reported anywhere? Demographics in each country? Demographics by Global North / Global South?
- d. Did random assignment work appropriately across the three technology conditions?
- e. Again, I don't know the norms for this journal, or area more broadly, but, personally, I would like much more detail on the methods in the main text. What are the survey instruments? What are the instructions? What are the technology descriptions? How did translations work? And so on. It feels "impossible" to evaluate these results without going to the Appendices. That seems problematic.
- f. Will data be publicly available for verification and replication? (When and under what conditions?)

VII. Language versus Location.

Translation: Was there any piloting done to compare descriptions across languages using respondents who speak both languages? Obviously this couldn't be done for every combination, but perhaps using English as a standard would provide valuable information. (I should acknowledge the ethnocentric suggestion; but it seems to be the most convenient and frequently-available comparison.) I acknowledge that this is a very difficult task, but anything that can increase confidence that we are seeing effects of location rather than survey language could be quite valuable.

VIII. Minor questions.

1. Breaking up the paragraph that begins on 238 could be helpful for many readers. A few additional sentences that explain some of the connections between presuming and domestic abuse and between natural gas production and prostitution could be useful.
2. Why two panels in Figure 1?
3. Line 696 - "The findings involving age are clearly relevant for policy and developing communication strategies." What is the policy relevance?

SUMMARY:

As I stated at the outset, these data are likely to be a major source of gravity with respect to any subsequent discussion about perceptions of geoengineering options. Such discussions are happening across fields and disciplines and well beyond academia. Not only will subsequent surveys and experiments have to 'answer to' (i.e., contextualize themselves with respect to) these results; but decisions about where to invest resources in technology development and governance might be influenced by these results. It is thus critically important that readers can clearly understand the contributions and limits of the current findings; and have every opportunity to calibrate their confidence in the findings appropriately.

Reviewer #2 (Remarks to the Author):

Dear Authors

Thank-you for the opportunity to review COMMSENV-23-1535-T. In this manuscript, the authors describe a multi-country survey to describe the connection between demographic aspects and support for climate mitigation technologies (e.g., BECCS, DAC, biochar). Overall, the MS was well-written, but the Supplementary Materials contain superfluous information and an overly detailed description of the methods. Overall, I commend the authors for deploying a survey which encompasses both Global North and Global South countries, but do not believe this manuscript is suitable for publication in this journal. I encourage the authors to seek out a more suitable venue for publication, and perhaps adjust the scope of the MS to a briefer research note. In support of this recommendation, I provide detailed comments and suggestions below.

pp3 The authors note that Mahajan and colleagues warn that concerning solar radiation management techniques in particular, "there is surprisingly limited empirical support that preferences for solar geoengineering vary by gender."

The inclusion of this claim is confusing, as it appears that Mahajan et al find no difference in support by gender, and yet the authors seem to be using this claim to bolster their interest in exploring gender effects. If instead the authors are referencing this study to claim that there has been insufficient work on gender differences in support for climate mitigative technology – this should be made clearer.

pp3 The claim “Tentative conclusions from the few studies that have examined youth perspectives on geoengineering have noted that younger people tend to prioritize climate action more strongly, but also to more strongly emphasize the need for international cooperation and governance. It is also youth that are more likely to be on social media, a platform they can use to reach millions of other individuals when they discuss climate policy or technology.” is too descriptive, and doesn’t provide a clear rationale as to why the authors are interested in the youth perspective.

pp4 The authors write “Given substantial demographic shifts currently underway around the world...”. What are these demographic shifts?

pp4 The authors write “Understanding how such factors enhance social diversity in climate decision-making and environmental organizations may also help speed the development of innovative technological and policy solutions.” It is not clear what this sentence means, nor does it appear to relate to the focus of the study (on demographic differences in support for climate mitigation technologies).

pp4. Similarly for the sentence: “though understanding points of tension (and support) across different individuals can more broadly help to support more equitable and effective development and deployment processes.” Understanding points of tension within individuals does not seem to be within the scope of this study.

pp8. While certainly there are well-established gender differences, the claims that “Women are more at risk to technology abuse and even domestic violence pertaining to the adoption of smart homes, smart meters, household energy control systems and digitalization of energy practices. Natural gas extraction and shale gas production, considered a bridge to low-emissions economies by some, also perpetuates increased rates of prostitution, sexually transmitted diseases and stillbirths, and threats to food security which more severely threaten women.” are overly specific for the scope of this study. In other words, I am not challenging the veracity of the forms of violence faced by women, but the specificity of the geographic and social contexts which produce this violence is not matched by the scale at which this present survey is deployed.

pp12-13. Since this study spans the globe, it would be helpful to include data on the vulnerability of people living in poverty (vs those not living in poverty) in Global North countries as well. Further, are all comparisons illustrated in Figure 1 significant?

pp16, Results and Discussion. Given that there are multiple statistical comparisons made, there should be an appropriate adjustment of the significance level, i.e., Bonferroni corrections. Related, there are no details provided of the statistical analysis conducted, e.g., a U-value for the Mann-Whitney test, F and p values for the ANOVA tests, etc.

pp16, Section 5.1. While the grouping of SRM technologies makes sense, I am not clear on the rationale for the other two groupings (and as described in the Supplemental material). Why was Biochar and Enhanced Weathering grouped separate from Soil Carbon Sequestration? These seem like very similar processes, akin to the SRM grouping. Further, while DAC is a fully engineered technology, BECCS involves both natural processes (plant photosynthesis) as well as combustion and carbon storage. Thus, the third grouping seems rather arbitrary.

Further, it is not clear that a focus on 10 separate carbon mitigation technologies is warranted here. The authors do not justify why it was necessary to include all of these specific technologies in the survey, as opposed to investigating public perceptions of climate mitigation technologies more generally (pulling excess CO₂ from the atmosphere, reflecting solar radiation) vs efforts that prevent the release of carbon dioxide in the first place (e.g., decarbonization). Overall, the focus on these 10 different technologies makes for an overly complicated paper. I suggest that either the Introduction be reframed to clearly justify why it is necessary to gauge support for each of the 10 technologies, or to reframe the paper as suggested.

pp16, Section 5.1.1. Blue Carbon is misspelled, and there should be a comma between Blue Carbon and DACCS

pp17. Figure 3 is not referenced in the text, and it is not clear why there two versions of the same figure are included.

Overall, effect sizes reported in this study are small (further, it is not clear whether statistical significance has been achieved for the reported analyses, given that corrections for multiple comparisons were not mentioned). The exception may be differences in support for SRM technologies between Global North and Global South countries.

In both the Results and Discussion, and in the Conclusion, the authors focus more on describing the results rather than on highlighting how these results can inform our approach to addressing climate change as a global community, how they inform the communication best practices around carbon mitigation technologies, how they can provide insights for policy rollout, etc., etc. This should be the most important part of the Discussion/Conclusion for this paper. Additionally, if there are indeed statistically significant demographic and intersectional identity effects, what might explain these particular results? In light of the detailed literature review provided in the Introduction on these demographic aspects, there are few new insights provided from this globally deployed survey (as currently described).

Finally, I am curious why the authors did not include values and/or political orientation (not specific parties, as that would be inappropriate – but whether an individual is more liberal or conservative), given that these are significant explanatory variables in belief in/concern about climate change, support for climate change policies, etc., I am surprised that they weren't included in this study.

Communications Earth & Environment is committed to improving transparency in authorship. As part of our efforts in this direction, we are now requesting that all authors identified as 'corresponding author' create and link their Open Researcher and Contributor Identifier (ORCID) with their account on the Manuscript Tracking System prior to acceptance. ORCID helps the scientific community achieve unambiguous attribution of all scholarly contributions. You can create and link your ORCID from the home page of the Manuscript Tracking System by clicking on 'Modify my Springer Nature account' and following the instructions in the link below. Please also inform all co-authors that they can add their ORCIDs to their accounts and that they must do so prior to acceptance.

Author Rebuttal letter: The author's response to these comments can be found at the end of this file.

Version 1:

Decision Letter:

Dear Professor Sovacool,

Your revised manuscript titled "The demographics of public preferences for ten carbon removal and solar geoengineering interventions" has now been seen by 2 reviewers, whose comments are appended below. You will see that they continue to find your work potentially interesting. However, substantial issues regarding the methodology and framing of the results remain, and we are not entirely sure whether you are willing and able to address these concerns. If you believe you can meet our editorial thresholds as outlined below, we would be interested in considering a revised version that fully addresses these serious concerns.

We hope you will find the reviewers' comments useful as you decide how to proceed. Specifically, for publication in Communications Earth & Environment to be appropriate, we would need you to:

- * provide a compelling statistical analysis that supports your main conclusions regarding the demographic distribution of support for various carbon removal and geoengineering interventions; this would have to include full justification of the grouping of technologies and the statistical methods employed (as requested by both reviewers).
- * include appropriate, clear and transparent caveats in the main manuscript, where limitations of the methodology lead to uncertainties or possible alternative explanations

If you choose to take up this option, please either highlight all changes in the manuscript text file, or provide a list of the changes to the manuscript with your responses to the reviewers.

Please use the following link to submit your revised manuscript, point-by-point response to the reviewers' comments with a list of your changes to the manuscript text (which should be in a separate document to any cover letter), a tracked-changes version of the manuscript (as a PDF file) and any completed checklist:

Link Redacted

Please do not hesitate to contact us if you have any questions or would like to discuss the required revisions further. Thank you for the opportunity to review your work.

Best regards,

Heike Langenberg, PhD
Chief Editor
Communications Earth & Environment

EDITORIAL POLICIES AND FORMAT

If you decide to resubmit your paper, please ensure that your manuscript complies with our editorial policies and complete and upload the checklist below as a Related Manuscript file type with the revised article:

Editorial Policy Policy requirements
(Download the link to your computer as a PDF.)

- Behavioural and social science
- Ecological, evolutionary & environmental sciences
- Life sciences

<https://www.nature.com/documents/nr-reporting-summary.zip>

For your information, you can find some guidance regarding format requirements summarized on the following checklist:

(<https://www.nature.com/documents/commsj-phys-style-formatting-checklist-article.pdf>) and formatting guide

(<https://www.nature.com/documents/commsj-phys-style-formatting-guide-accept.pdf>).

REVIEWER COMMENTS:

Reviewer #2 (Remarks to the Author):

I really admire this research effort. I appreciate the improvements made to the revision on many dimensions. I believe we can learn a lot from these data. However, in my opinion, the current writeup remains problematic.

1. It is not at all clear if these results generalize beyond these specific descriptions of the different technologies, and, in my view, this fundamental issue is not addressed clearly in the manuscript. The core question here is how different people (specifically different demographic groups of people) evaluate a stimulus. But the stimulus here is one authorship team's descriptions of the technologies, not the technologies themselves. There is no empirical reason to believe that the stimulus is representative of the class of stimuli which it is supposed to stand for. I suspect that I (or anyone) could write different (reasonable) descriptions of the same technologies and find very different results. Of course, then reasonable people could and should debate which descriptions are more reasonable / appropriate / representative etc. But any reader should have serious questions about how important the specific details of the descriptions are, and my concern is that the writeup is not clear enough about this limitation of the work. I fear that many readers (scientists, media, and public) will overinterpret based on the current discussion. This concern is exacerbated by frequent declarations about the policy relevance of these results. (For example, if we include even minimal information about risks and uncertainties, we might see very different patterns and levels of support.) I believe strongly that this limitation of the research design should be front and center in every major section of the paper (abstract, introduction, methods, discussion). I am not suggesting that the researchers can or should go back and re-run with multiple descriptions – that is impractical at this stage. It was a reasonable decision for them to make for all of the reasons they elaborate in the letter. But, in my opinion, it is essential that it be discussed openly and clearly as a methodological choice that limits interpretation. Readers might be somewhat reluctant to accept that there is “substantial policy relevance” from one result based on one description.

The ambiguous sentence that, “This complicates [interpretation] in some ways...” is not adequate in my view.

2. I remain concerned that the conclusions about specific technologies may not be warranted given the nested research design. Individual technologies were completed nested within groups of technologies. (Again, this was a reasonable decision based on response fatigue, etc.) But treating them as independent in analysis and conclusions therefore seems inappropriate. Maybe I am off base here, but – given the likely influence of this important work – I highly recommend that an additional reviewer who specializes in data analysis be consulted in the process before publication (if one has not been already).

3. Ignoring the nesting problem, it is not clear to me that the results merit the statement that there is “support on average” for the technologies. If I am reading correctly, few if any of these means are significantly different from the scale midpoint. Was this tested? For many of the technologies, it seems just as accurate to assert that the means indicate “neither rejecting nor supporting on average.”

4. I am confused by the paragraph (p. 16 on my view) discussing “priming.” How can it be that the materials avoid priming participants with risks versus benefits and also that the descriptions offer a balanced presentation of risks and benefits? I recognize and appreciate that this was added in an effort to address one of my questions, but I have to admit that I am confused by the idea here. In any event, I highly recommend – for this reason and comment (1) above – that the descriptions be included in the main text instead of requiring readers to go to SOM. It seems that they could fit in a table without taking up too much journal space.

I expect this work to have substantial influence. I hope that the published version will be written up in the most transparent and balanced form possible to help maximize learning and minimize misinterpretation. I believe that the authors share this goal and I hope these comments help push them further in this direction.

Reviewer #3 (Remarks to the Author):

Dear Authors

Thank-you for the opportunity to review the revised MS COMMSENV-23-153A. As before, I commend the authors for deploying a survey which encompasses both Global North and Global South countries, but still do not believe this manuscript is suitable for publication in *Nature Communications: Earth & Environment*. I encourage the authors to seek out a more suitable venue for publication, and/or adjust the scope of the MS to a briefer and more focused research note. In support of this recommendation, I reiterate my main concerns below.

The authors present a long literature review detailing vulnerabilities to, and concern with, climate change across demographic and socioeconomic categories (and resulting intersectional identities). However, there remains scant theoretical justification for why one may expect difference in support for 10 different forms of geoengineering and carbon capture technologies across these different sociodemographic groupings. Overall, the introduction provides insufficient justification for the study design, analyses, and interpretation of results. The introduction should be rewritten to provide a clear, logical, and succinct justification for why we should expect demographic and socioeconomic differences in support for specific geoengineering and carbon capture technologies. For example, I believe that there are interesting reasons to focus on differences between youth and adults in support for geoengineering and carbon removal technologies. However, the justification for the focus on youth in this study is still insufficient and overly general. To say that “Youth are more vulnerable to the duration and severity of impacts of climate change than adults, given that they will generally live longer (incurring more exposure) but also presently have physiological factors (such as smaller lungs and less developed immune systems) that make particular impacts such as air pollution or heat stress more extreme.” certainly justifies a focus on risk perceptions and, perhaps, support for actions to mitigate climate, but not on why youth might differ in their support for specific types of geoengineering and carbon capture technologies. Similarly, while the authors mention gendered differences in support for, perceptions of, and use of technology, I don’t see a clear connection to the types and groupings of the technologies used in this study.

Related, the MS still lacks sufficient justification as to why it is necessary to conduct a survey examining 10 specific types of geoengineering and carbon capture technologies on a global scale. Further, these technologies were organized into three groups SRM (stratospheric aerosol injection, marine cloud brightening, space-based geoengineering); ecosystem-based CDR (afforestation and reforestation, soil carbon sequestration, marine biomass and blue carbon); engineered CDR (direct air capture with carbon storage (DACCS), bioenergy with carbon capture and storage (BECCS), enhanced weathering, biochar). I mentioned in my initial review that the groupings were inadequately justified, and I reiterate that concern again. The authors point to three studies to justify their groupings (Morrow et al. (2020), Low et al. (2022), and Sovacool et al. (2022)). However, these studies do not appear to match the carbon removal technology groupings used in the present study; these published studies used, for example, “Nature-based carbon dioxide removal (e. g., afforestation and reforestation, biochar, soil sequestration, blue carbon and seagrass, ecosystem restoration)” and Engineered carbon dioxide removal (e. g., carbon capture and utilization and storage, bioenergy with carbon capture and storage, ocean iron fertilization, enhanced weathering, direct air capture) or they used “Nature-based, Hybrid and Engineered carbon removal”. Further, many of these technologies are unproven and have had limited application and discussion outside of policy and academic circles. I thus have concerns with what participants (particularly those with limited literacy and education) might be responding to in these scenarios given their low to no exposure outside of this survey setting (i.e., survey response effects).

I remain unsatisfied with the reporting of statistical analysis. The authors have provided insufficient justification for the tests used, and in the reporting of results. Specifically, why were both parametric and nonparametric tests used? Given the lack of justification for a focus on intersectional identity effects and support for geoengineering and carbon capture technologies, why were three-way ANOVAs conducted (as opposed so one-way, or two-way?)? How was education level (and other confounding variables) controlled for in these tests? Overall, there remains insufficient detailing of the analyses used beyond the reporting of effect sizes.

Finally, while a focus on poverty is interesting, I am unclear as to why the authors used a poverty threshold established for higher income countries. Further, given that most countries in the study had zero participants below the poverty level, do

country level effects not confound the effect of poverty? Overall, only 3.3% of survey respondents could be categorized as living in poverty, and these came from 8 countries (as six of these eight countries were from the Global South, that is also a confounding factor). Further, the authors explanation for how the poverty threshold was arrived at is confusing as written.

Communications Earth & Environment is committed to improving transparency in authorship. As part of our efforts in this direction, we are now requesting that all authors identified as 'corresponding author' create and link their Open Researcher and Contributor Identifier (ORCID) with their account on the Manuscript Tracking System prior to acceptance. ORCID helps the scientific community achieve unambiguous attribution of all scholarly contributions. You can create and link your ORCID from the home page of the Manuscript Tracking System by clicking on 'Modify my Springer Nature account' and following the instructions in the link below. Please also inform all co-authors that they can add their ORCIDs to their accounts and that they must do so prior to acceptance.

If you experience problems in linking your ORCID, please contact the Platform Support Helpdesk.

Author Rebuttal letter: The author's response to these comments can be found at the end of this file.

Version 2:

Decision Letter:

Dear Professor Sovacool,

Your manuscript titled "The demographics of public preferences for ten carbon removal and solar geoengineering interventions in 30 countries" has now been seen by 1 original reviewer and 1 adjudicator, and we include their comments at the end of this message. They find your work of interest, but some important points are raised. We are interested in the possibility of publishing your study in Communications Earth & Environment but would like to consider your responses to these concerns and assess a revised manuscript before we make a final decision on publication.

We therefore invite you to revise and resubmit your manuscript, along with a point-by-point response that considers the points raised. Please highlight all changes in the manuscript text file. In particular, for publication in Communications Earth & Environment, we request that you (1) address the nesting approach as suggested by reviewer #1 to avoid a bias, (2) tone down the results related to the poverty and emphasize the results related to youth and gender and (3) clarify how your results differ from the recently published studies.

Please submit your point-by-point responses as a separate file, distinct from your cover letter where you can add responses to the Editors' comments that you do not want to be made available to the reviewers. Word files are preferred.

Important: The response to reviewers must not include any figures, tables or graphs. If you wish to respond to the reviewer reports with additional data in one of these formats, please add them to the main article or Supplementary Information, and refer to them in the rebuttal. Due to current technical limitations, any figures, tables, or graphs embedded in your rebuttal will not be included in the peer review file, if published.

Please use the following link to submit your revised manuscript, point-by-point response to the referees' comments (which should be in a separate document to any cover letter), a tracked-changes version of the manuscript (as a PDF file) and the completed checklist:

Link Redacted

We hope to receive your revised paper within six weeks; please let us know if you aren't able to submit it within this time so that we can discuss how best to proceed. If we don't hear from you, and the revision process takes significantly longer, we may close your file. In this event, we will still be happy to reconsider your paper at a later date, as long as nothing similar has been accepted for publication at Communications Earth & Environment or published elsewhere in the meantime.

Please do not hesitate to contact us if you have any questions or would like to discuss these revisions further. We look forward to seeing the revised manuscript and thank you for the opportunity to review your work.

Best regards,

Martina Grecequet, PhD
Associate Editor,
Communications Earth & Environment
@CommsEarth

EDITORIAL POLICIES AND FORMATTING

Editorial Policy: [Policy requirements](https://www.nature.com/documents/nr-editorial-policy-checklist.pdf) (Download the link to your computer as a PDF.)

- Behavioural and social science
- Ecological, evolutionary & environmental sciences
- Life sciences

<https://www.nature.com/documents/nr-reporting-summary.zip>

Furthermore, please align your manuscript with our format requirements, which are summarized on the following checklist: [Communications Earth & Environment formatting checklist](https://www.nature.com/documents/commsj-phys-style-formatting-checklist-article.pdf)

and also in our style and formatting guide [Communications Earth & Environment formatting guide](https://www.nature.com/documents/commsj-phys-style-formatting-guide-accept.pdf).

*** DATA: Communications Earth & Environment endorses the principles of the Enabling FAIR data project (<http://www.copdess.org/enabling-fair-data-project/>). We ask authors to make the data that support their conclusions available in permanent, publically accessible data repositories. (Please contact the editor if you are unable to make your data available).

All Communications Earth & Environment manuscripts must include a section titled "Data Availability" at the end of the Methods section or main text (if no Methods). More information on this policy, is available at <http://www.nature.com/authors/policies/data/data-availability-statements-data-citations.pdf>.

If a community resource is unavailable, data can be submitted to generalist repositories such as [figshare](https://figshare.com/) or [Dryad Digital Repository](http://datadryad.org/). Please provide a unique identifier for the data (for example a DOI or a permanent URL) in the data availability statement, if possible. If the repository does not provide identifiers, we encourage authors to supply the search terms that will return the data. For data that have been obtained from publically available sources, please provide a URL and the specific data product name in the data availability statement. Data with a DOI should be further cited in the methods reference section.

Please refer to our data policies at <http://www.nature.com/authors/policies/availability.html>

REVIEWER COMMENTS:

Reviewer #2 (Remarks to the Author):

1. This revision has again improved on multiple dimensions. The handling of “descriptions” of technology at appropriate sections in the manuscript is clear and straightforward and I think does a great service to helping readers interpret the findings appropriately. Improved precision in interpreting the mean values is similarly helpful.

2. I do not yet understand how the nesting issue has been addressed (i.e., implied cross-group comparisons based on fully confounded group presentation).

Imagine we want to know, How healthy is Coca Cola?

Imagine Survey A: How healthy or unhealthy are each of the following beverages: Coffee, Tea, Matcha, Coca Cola.

Imagine Survey B: How healthy or unhealthy are each of the following beverages: Frappuccino, Milkshake, Coca Cola.

It is possible that Coca Cola comes out looking worse in the first group than the second group based on its relative standing in the group.

Now imagine you want to know how people rank the healthiness of the following beverages Coffee, Tea, Matcha, Frappuccino, Milkshake, and Coca Cola. An experiment that seeks to rank Coca Cola based on the combined results of the following two groups {Coffee, Tea, Matcha}, {Frappuccino, Milkshake, Coca Cola} MIGHT produce a different ranking than an experiment that seeks to do so based on the groups {Coffee, Tea, Matcha, Coca Cola}, {Frappuccino, Milkshake}. Neither the analyses nor the interpretations acknowledge this potential source of bias. If Van den Brakel (2019) puts this issue to rest, I do not understand how and explanation might be appropriate. The discussions on pp.18-19 appropriately address potential “extraneous factors” that reside within the participants (or different groups of participants), but not extraneous factors attributable to stimulus grouping. Steps like breaking Table 1 into separate tables or panels defined by the four groups would, in my opinion, help to discourage / decrease comparisons potentially inappropriate cross-group comparisons. Explicitly acknowledging this issue would be even more helpful to readers.

3. I do not understand the following argument:

“Nevertheless, as is necessarily the case when providing information, had different information been provided, respondent evaluations of the interventions could have differed. We however note that, having ruled out any (unintended) significant differences among the groups randomly assigned to the technology categories, we cannot identify material reasons for why such differences should have occurred.”

How does the second sentence bear on the first?

4. I do not know the journal’s policy, but I wonder if a final version of this article should address how it differs from and builds on the now-published work by Baum and colleagues (2024) in Nature Communications.

5. There seems to continue to be conflicting information about whether this work should be viewed as an experiment or not. The response letter takes a strong stance that it is not an experiment, but then refers to it as an experiment a few lines down. The manuscript talks about embedded experiments. I don’t have a strong stance, but it is a little confusing.

6. I am confused by this sentence:

“For income levels and poverty, the most basic observation is also one of the most striking – that those in poverty were nearly universally more supportive of climate interventions of various types, compared to people not in poverty”

What does “nearly universally” mean here? The universe of technologies? Of participants? It seems like it could easily be misinterpreted either way. If it is referring to distributions of opinion among those in poverty versus those not in poverty, is it really near universal? Do the distributions not overlap substantially?

Reviewer #4 (Remarks to the Author):

This article contributes to a small but growing literature on perceptions of carbon dioxide removal and solar radiation management outside of a few well-studied countries in the Global North. The scope of the work is impressive, and the methods rigorous. I find the authors’ responses to reviewer comments adequate, barring the question of the poverty level cut-off—which, as I see it, is central to their analysis, and thus warrants revision or further justification.

While I appreciate the authors’ acknowledgement (in the rebuttal and in the manuscript) that poverty must necessarily be flattened to be compared in an analysis like this, I am uncomfortable with the extent to which the authors flatten it—especially given that the authors present the role of poverty in perceptions of climate interventions as one of their primary contributions. Their current measure is a poverty threshold (even if a conservative one) for Global North countries. As reviewer #3 notes, there are two problems with this. The first is that this threshold ends up being artificially high for

developing and lower-income countries, and the second is that poverty in Global North countries is poorly represented given the higher [universal] threshold (and, presumably, sampling). I am not convinced that the findings in this paper demonstrate something about the role poverty plays in perceptions of climate interventions that is different from the finding already reported in Baum et al. (2024) in Nature Communications that publics in the Global South show significantly greater support for climate intervention strategies than those in the Global North. While the findings on gender and youth are interesting, if these are to become the central tenets of the paper, the authors should better explain the importance/implications of these findings to make up for the loss of the poverty metric. It is also unclear to me why education wasn't included in the battery of covariates, as that (along with political orientation and, depending on the national context, race and/or ethnicity) is very often used in analyses like these, as it often shapes perceptions of climate issues.

In conclusion, while I would like to see this article published, I am not convinced by the poverty analysis and fear that it may be a reiteration of an analysis that has already been published with this dataset.

Communications Earth & Environment is committed to improving transparency in authorship. As part of our efforts in this direction, we are now requesting that all authors identified as 'corresponding author' create and link their Open Researcher and Contributor Identifier (ORCID) with their account on the Manuscript Tracking System prior to acceptance. ORCID helps the scientific community achieve unambiguous attribution of all scholarly contributions. You can create and link your ORCID from the home page of the Manuscript Tracking System by clicking on 'Modify my Springer Nature account' and following the instructions in the link below. Please also inform all co-authors that they can add their ORCIDs to their accounts and that they must do so prior to acceptance.

Author Rebuttal letter: The author's response to these comments can be found at the end of this file.

Version 3:

Decision Letter:

Dear Professor Sovacool,

Thank you for submitting your manuscript titled "The demographics of public preferences for ten carbon removal and solar geoengineering interventions in 30 countries". We are delighted to say that we are happy, in principle, to publish a suitably revised version in Communications Earth & Environment.

We therefore invite you to edit your manuscript to comply with our format requirements and to maximise the accessibility and therefore the impact of your work.

EDITORIAL REQUESTS:

****Please take care to match our formatting and policy requirements. We will check revised manuscript and return manuscripts that do not comply. Such requests will lead to delays. ****

Please outline your response to each request in the right-hand column. Please upload the completed table with your manuscript files as a Related Manuscript file.

SUBMISSION INFORMATION:

In order to accept your paper, we require the files listed at the end of the Editorial Requests Table; the list of required files is also available at <https://www.nature.com/documents/commsj-file-checklist.pdf> .

OPEN ACCESS:

Communications Earth & Environment is a fully open access journal. Articles are made freely accessible on publication. For further information about article processing charges, open access funding, and advice and support from Nature Research, please visit <https://www.nature.com/commsenv/open-access>

Link Redacted

Best regards,

Martina Grecequet, PhD
Associate Editor,
Communications Earth & Environment
@CommsEarth

Editor and reviewer comments	Response from the authors
Summary of revisions	We appreciate the opportunity to revise and improve the paper and have made multiple substantive revisions in response to the referee comments. We believe the revised manuscript fully meets the first criterion about insights into the demographic dimensions of public preferences for the 10 climate intervention technologies examined. To clarify, we would not classify our study’s research design as experimental or quasi-experimental. It has no control group, nor any sort of choice experiment. (We considered this, but ruled it out due to lack of our own expertise and lack of resources). Instead, we would classify our study as falling into the “Surveys and quantitative data collection” category (according to https://www.sciencedirect.com/science/article/pii/S2214629618307230), which is distinct and has different codes of practice, but is very well established in the social sciences literature. That said, we have nevertheless better justified our variables and selection of technologies in our revision, and provided more details about significance tests run.
Reviewer #1: This is a tremendously ambitious effort to collect some of the most broadly representative data anyone has seen on perceptions of climate-engineering technology. This dataset is poised to become a focal point in all subsequent discussions of “social license” (or other preferred term). The initial report therefore has a tremendous responsibility, in my opinion, to be crystal clear about what conclusions it does versus does not support. It must therefore be held to the highest standards of rigor and transparency. While acknowledging the very high value of this effort at the outset, I elaborate on my key concerns along each of	We genuinely appreciate your very comprehensive, thoughtful, and reflexive assessment. As you guessed, none of the authors come from the fields of behavioral science, environment and behavior, or applied psychology. Our expertise is instead rooted in energy policy, climate policy, energy social science and science and technology studies. We recognize that standards of rigor and appropriate design do differ across these disciplines, and appreciate you not judging us solely on standards from your own discipline(s).

these dimensions below. My central concern is that so much hinges on one particular description of each technology and one particular judgment experience (within a cluster of other judgments). I think many readers will be inclined to think that there is some level of fundamental truth here because the samples are so big and so broad – a true strength! – but will fail to appreciate that these methods can say little about whether any technology judgment is robust to the exact description or order in which it came. This is not a fatal flaw but must, in my opinion, be discussed as a major limitation. My other major concern is that the reporting of methods, analyses, and results would be considered insufficient for behavioral-science outlets that I am familiar with. Before I elaborate on these concerns, I also discuss important questions about the scope and contribution of the current subset of analyses.	Also, as explained to the editor above, we would not classify our research design as experimental or quasi-experimental, it is instead a more simple (or straightforward) representative survey design, which has different criteria and standards of rigor than an experimental design, see here for more https://www.sciencedirect.com/science/article/pii/S2214629618307230. It is widely used in the social sciences, perhaps even more widely used than experimental designs. That said, we did put much care and effort into our research design, to the point where Reviewer 2 even criticized is for “an overly detailed description of the methods.” (This also clearly evidences the differences in norms and standards across disciplines that we mentioned above). In our revision, we’ve gone even further to add more details, and make explicit more of the assumptions and limitations in our approach. We hope this ensures that readers are impressed not just with our big and broad sample sizes, but also the thought we put into the design of the survey instrument.
I would also contextualize my comments from the outset by saying that I do not work with such big datasets in my own work. I trust that colleagues with more statistical expertise will be consulted on the best approaches for modeling these results; especially in light of the choice to use a mixed design (with specific technologies grouped into technology categories).	Indeed, we should mention that one of our coauthors (Baum) was hired explicitly because of their training and expertise in both survey design and multi-technology surveying and assessment, as well as mixed methods research designs.
I. Scope and Contribution. I work mostly in a field where the prevailing norm is multiple studies per paper rather than multiple papers per study. Therefore, I do not know how Authors and Editors should negotiate the bounds of a single report based on such a large study. I will merely note that I think there are clear opportunities to learn substantially more about the stated research question(s) in this case. The authors have done an excellent job of pitching the importance of demographic factors	Again, thank you for being so reflexive on this point, and non-dogmatic. On the point about number of papers per project/data source (like a survey), this is something many scholars grapple with. In our case, the survey is one of the flagship outputs for a 6-year ERC grant that took us 1.5 years to design, translate, test, implement, analyze, and write up, at a cost

in this societal discussion; but even in their own introduction to these issues, it is clear that the demographics are interesting insofar as they represent typical psychological concerns, needs, or access to information. Given the scope of the data-collection effort, it seems fair that a research team would reserve other questions for other reports. However, in my reading, we could potentially learn a lot more about why the demographic predictors show what they show by beginning to connect some of the psychological variables. For example, we might be able to learn a lot more about why age shows the patterns it shows by examining whether variables such as trust in science, sources of information, affect, or perceived risk account for the age finding. Similarly, we might learn a lot more about why poverty shows the patterns it shows by examining appropriate covariates as potential mediators. I acknowledge how subjective this suggestion might be – others who are more familiar with the norms here may feel quite differently – but it seems to me that the fundamental reason for identifying demographic predictors is because those are proxies for what information different groups tend to think of (and how they think about it) when they evaluate these technologies.	of about US\$500,000. To write only one paper on this rich, original dataset seemed to us a waste of resources but also poor practice for the advancement of science. Also, the scale, scope, and cost of our survey endeavor is closer to that of the Eurobarometer or OECD EPIC surveys from which researchers often write dozens to hundreds of papers. So we feel it entirely appropriate to write a handful of different papers based on the survey as long as they explore different results, are framed for different journals, and are entirely original (having less than 1% overlap with each other), which is what we have done. As to the point about norms, even though they are not in our field, we did consider particular frameworks such as the Norm Activation Model or Value Belief Norm Theory at the start of the process when we chose the variables for our survey, but decided against them based on overly constraining our focus and results. And, on us not being trained in behavioral science. That said, we agree that “a fundamental reason for identifying demographic predictors is because those are proxies for what information different groups tend to think of (and how they think about it) when they evaluate these technologies”, and now mention this in the Introduction as another value to the survey, with this as support: Beiser-McGrath, L. F., & Huber, R. A. (2018). Assessing the relative importance of psychological and demographic factors for predicting climate and environmental attitudes. Climatic change, 149, 335-347.
II. Technologies versus Descriptions of Technologies. I think it could set a very bad precedent to suggest in any form that this work allows comparisons or rankings of preferences between TECHNOLOGIES, per se.	Fair point. Firstly, let us clarify that these sorts of Nature journals prefer that much of the methodological detail stays in Supplemental Materials, that is just their style, and it’s why our SM file is so long. However, we have moved some of this detail into the paper itself, to better justify our

Research from many fields, disciplines, and traditions shows clearly that numerous details about the description can change people’s perceptions (ranging, most relevantly, from risk information to specific word choices to comparisons with and contrasts to other technologies). The authors are careful not to offer direct statistical comparisons, nor interpretative statements that would advance this narrative; but they also state that the ability to compare across technologies is a potential strength and contribution of the work, and the figure design seems to invite these rankings. There are some very thoughtful discussions about this point in the Supplemental materials. I would suggest moving extended versions of these discussions to the main text to help readers avoid drawing overgeneralizations.	choices and also flag for readers where they can learn more in our SM. Secondly, we believe the multi-technology focus of the survey is one of its core strengths, almost equal in importance to the survey being run in multiple countries as well. To better justify this, we have added this paragraph to the manuscript, with reference to a 2024 study arguing in favor of focusing on so many technologies: “we examine all ten [climate intervention technologies] together as an integrated portfolio because this is how they may be synergistically deployed together as part of a future climate policy package, and because both suites of carbon removal and solar geoengineering technologies are shaping climate governance and mirrors the policymaking dilemma of choosing options with limited resources and uncertainty,” with this for support: Benjamin K. Sovacool, Chad M. Baum, Roberto Cantoni & Sean Low (2024, in press) Actors, legitimacy, and governance challenges facing negative emissions and solar geoengineering technologies, Environmental Politics, DOI: 10.1080/09644016.2023.2210464
For example: “Furthermore, due to the novelty of the technologies, the determination was made to avoid ‘priming’ participants by descriptions that overly focused on risks versus benefits, or vice versa, and as much as possible to talk about how technologies would work rather than what might go wrong, especially where significant uncertainty still prevailed.” It seems that this needs to go front and center in main text. There is no “neutral,” so, of course, researchers have to make a choice. But it should be acknowledged that they are studying responses to [the chosen descriptions of the technologies] not	Excellent suggestion, agreed on two counts. Firstly, we have added this to the description of methods and removed it from the SM: “Due to the novelty of the technologies, the determination was made to avoid ‘priming’ participants by descriptions that overly focused on risks versus benefits, or vice versa, and as much as possible to talk about how technologies would work rather than what might go wrong, especially where significant uncertainty still prevailed.”

responses to [the technologies themselves]. In other words, this single study, using one description per technology, is not able to do much to increase confidence that the absolute levels of acceptance would generalize beyond the specific stimulus used in the current study. It also seems critical to me that technology descriptions (or, at the very least, representative examples) should appear in the main text.	Secondly, we have added short technology descriptions to the Introduction, and also added the 10 specific technologies covered by the study to the Abstract for good measure.
Another important point is provided later in the supplementals: “Probably the clearest example is ‘stratospheric aerosol injection’, given that this text is among the shortest and does not place substantial emphasis on possible impacts on the biosphere and weather patterns.... Still, we acknowledge that, by trying to provide a balanced presentation of risks and benefits and to provide participants broad information from which they could draw their own inferences, in this case, it may have unintentionally resulted in an insufficiently comprehensive illustration of potential risks.” These caveats seem critical to address head on in the main text.	We’ve met the referee halfway here—we have not included the full paragraph into the main text due to length considerations, but put a truncated version there as a sentence, noting that “Even though our technology descriptions (shown in Supplemental Materials) offered a balanced presentation of risks and benefits, due to uncertainty or lack of knowledge, respondents may nevertheless err on the side of overvaluing risks instead of benefits.”
I would also suggest that categories of technologies be presented in separate figures or panels to further discourage direct comparisons given the countless subjective choices that limit direct comparison.	We very much appreciate the need to be as clear as possible in all data visualisation. It seems the suggestion here is for separate figures each for solar radiation management, then for each of the two types of carbon dioxide removal. Whilst we do agree that this might help clarify the differences between the three groups, it might also mask (or at least make difficult for readers to deduce) the similarities between technologies that lie in different groups (e.g., solar radiation management and carbon dioxide removal group 2 – compared to carbon dioxide removal group 1 – in figures 3, 4, 5, and 6. Ultimately, it is clear there are differences between the three groups, but also that carbon dioxide removal group 1 is something of an outlier of its own, which is important for interpreting the overall results. A different, but still relevant, consideration

	is that the current manuscript already includes a substantial number of figures (10). There are an additional 16 figures in the supplementary materials. We believe it would become overwhelming and distracting to add a minimum of eight more figures to the main text. Indeed, we doubt a standard full length journal article would be able to accommodate this magnitude of data visualisation. We are certainly happy to provide additional figures in the supplemental online materials. We feel, however, that the current figures in the main text strike the right balance between appropriate magnitude of data presented and need for detail between technologies.
III. Experimental Design. Treating the ten technologies as independent items (conceptually and statistically) does not seem appropriate given the experimental design. As explained in the Supplemental materials, the authors were sensitive to the possibility of contrast or assimilation effects – they refer to the general possibility as anchoring – but then, as far as I can tell, they treat tests associated with each technology as independent from one another.	See above, we wouldn't consider our design experimental at all, and we also believe that focusing on differences between all ten technologies independently is a crucial advantage to the analysis. We certainly see some clustering amongst technologies that were presented to the same respondents (e.g., in figures 3, 4, 5, and 6), but it is difficult to deduce whether these effects are due to method effects or the fact that related technologies might engender similar attitudinal responses. We have added some additional text to the limitations sub-section at the end of the methods to clarify the non-independence of the technologies, clustered within the three groups.
It seems that most respondents are not very knowledgeable about the technologies. Therefore, they are likely to be looking for contextual cues. These cues might come from comparisons between technologies (As but one example: Marine cloud brightening might look less concerning /more favorable after thinking about SAI than after thinking about BECCS); and these cues might also come from generalizations across the category (for example: my intuition is that the analogous graphics for all three SRM	The reviewer raises a valid concern. Beyond the aforementioned additions to the limitations area of the methods section, we have also inserted text in the conclusions section to indicate that results may have varied had different combinations of technologies been presented. Nevertheless, the important patterns that we evidence about relationships between views of these technologies and sociodemographic characteristics transcend the groupings. Therefore, we see reason to believe that the

technologies might suggest that they are basically the same but at higher altitudes). This calls into question the independence of the specific judgements; and potentially limits the external generalizability of any given result.	findings are robust and not merely method effects (again, also acknowledging the random assignment to the information conditions).
A conservative approach might be to treat the three or four items as multiple items from a scale, leaving three categories of responses. Do measures of reliability support this approach? At the very least, order effects should be explored carefully.	Within each technology category, technologies were always presented in a randomised order to avoid learning effects or other related biases. There was some natural variation in number of participants viewing a technology first versus second or third (or fourth), owing to inconsistencies around which participants dropped out or were excluded. Such differences were, however, generally in the tens (out of more than three thousand for the SRM and ecosystems-based groups and around twenty-five hundred for engineered CDR). Regarding the suggestion of checks on ordering effects, we concur that this would be possible. This is why we felt it was important to counterbalance the order of presentation – so that any such biasing effects are smoothed out across participants. We conducted a series of one-way ANOVA analyses, considering whether the order in which technologies were presented influenced levels of support. We find no evidence of any ordering effects for the following technologies: stratospheric aerosol injection; marine cloud brightening; blue carbon and marine biomass; DACCS; and BECCS. There is evidence that support is significantly higher among those participants who read about space-based geoengineering and enhanced weathering earlier in the sequence. The opposite was true for afforestation and reforestation, and soil carbon sequestration. Conversely, there was greater support for biochar among participants reading about it first, closely followed by those reading about it last, with the lowest values among those viewing it in the middle.

	We report these results out of interest and to offer clarity to the reviewer. We take this, nevertheless, as justification for our decision to counterbalance the presentation order of the technologies – though it is notable that this did not have an effect for five of the ten technologies. In summary, there is no systematic pattern of ordering effects.
With respect to these issues, there was some ambiguity about many specific experimental procedures. Within technology category, were specific technologies presented in a randomized or counterbalanced order? Or were they always presented in the same order? If they were balanced in some way, then robustness checks could test whether all the effects hold when looking only at the first technology that a participant rated. This would be extremely valuable for any discussion that purports to compare the technologies head-to-head. If the orders within category were fixed, then some technology-to-technology comparisons are inevitably confounded with potential order effects.	Please see our response immediately above for our methodological approach and additional analysis in relation to order effects.
Do statistical analyses account for the technologies within groups? These aren't ten independent tests, they are three independent tests (SRM, Eco-CDR, Engineered-CDR). What are the alphas within categories? Would it make sense for the main analyses to be based on three categories rather than ten technologies?	We fully acknowledge that each respondent was only asked questions about one of the three categories. This was simply to avoid survey fatigue and satisficing. Due to the amount of background information we presented to the respondents on each technology, we needed to limit the number of technologies covered in each survey. The tradeoff was between breadth (number of technologies) and depth (amount of content on each technology). We could treat the SRM and the CDR1 and CDR2 items as three scales, but we do not feel that ten separate items is overly confusing to include in the data analysis, which would lead us to err on the side of more data rather than less. The reader can look for themselves at the data and judge the similarities, or lack thereof, between the

	items. For example, it is clear that support for soil carbon and for blue carbon are much more related to each other than either of these items is to afforestation. Combining all three of these items would mask this variation and nuance across the three technologies. In terms of the concern about non-independence of observations, we acknowledge this possibility and discuss it now in the expanded limitations section of the article (at the end of the methods). One problem with relying on alphas is that because questions about only one group of technologies were asked to each respondent, we cannot see how the respondents would have answered about the other technologies. For example, the alpha for SRM is 0.86, which is indeed high, but these three items might have pooled reasonably well with the CDR2 items also (we just cannot statistically test for this, because no one was asked both set of questions). Additionally, the alpha for CDR1 is high, at 0.80, but this masks the lack of similarity between afforestation and the other two items (i.e., the scale would be 0.83 if afforestation were deleted, noting that it is bringing the alpha down).
IV. Operationalization of Poverty. Were there any other measures of wealth or socioeconomic status used besides the very strict dichotomous World-Bank measure? It seems we could learn a lot more about the predictive power of wealth if additional analyses explore (a) continuous measures of wealth, and (b) relative measures of wealth. I am asking about (b) because subjective SES matters too. Income level X may make one relatively “rich” in one country and “poor” in another country; and it is possible that status offers unique predictive power beyond poverty. Perhaps a country-level percentile measure would be worth exploring.	Interesting comment. We were not entirely sure what is meant here by a country-level percentile measure. If the reviewer simply means the percentage of people in a given country at the poverty level, we would be concerned that a country-level variable in an individual-level analysis (i.e., one that does not employ hierarchical modelling) would lead to errors due to the non-independence of observations, if run alongside individual-level variables in the same model. That is, there would be problems due to treating a group-level variable as if it were data at the individual level. Also, we are not sure what the percentage would tell us. Would we be

	interested in whether poor people from less poor countries are different from poor people from more poor countries? We already look separately at the effects of being from the Global North vs Global South, and of poverty. However, to engage with this comment, we have presented a table with the percentage of the population of each country in our sample living below the national poverty line. This has been added to Table 2 (i.e., another column with the nationally-defined poverty percentage). This way we offer the data, but do not use it in any statistically-questionable analysis.
V. Composite Measure. I was surprised by the decision to collapse the three DV items, research / small-scale trial / broad deployment. I was even more surprised when I found the main result in the supplementals that apparently led to this decision: alpha values above 0.90. This seems like it might merit discussion in the main text, both because of the methodological importance and the potential context that it provides for interpretation. If participants are distinguishing between research and large-scale deployment at such low levels, this perhaps suggests that responses reflect something more like general attitudes than deliberative reflection.	Happy to clarify. We agree that the survey results reveal general attitudes towards each of the technologies. By having a dependent variable that is composed of multiple items, we believe this simply offers a more reliable measure of respondent attitudes towards the technologies.
Do people from some demographics distinguish between these three measures more than others do? If so, this would be an interesting finding in my opinion.	We have explored this question that the reviewer raises for effect of gender on support and effect of poverty on support. We present those figures at the end of this response letter. The results show very little variation in the influence of gender based on whether the dependent variable is support for research, support for small scale deployment, or support for broader deployment. We see the main effect of gender to the same degree as when using the combined support variable, but the pattern of the way in which gender

	affects support is broadly the same irrespective of which of the three sub-measures of support one examines. We find the same results for the effect of poverty on each of the three sub-measures of support. We could add these additional figures to the Supplemental Materials, but we have left them out for now, to avoid confusion in a document quite full of additional figures already (and due to the lack of any meaningful difference based on operationalisation of support being used).
VI. Reporting and Transparency: I am not familiar with this journal's reporting norms. As a true generalist journal, I imagine there is quite high variance. I was particularly surprised, however, that almost no descriptive statistics were reported in the main text or supplementals as far as I could tell. Readers cannot even find anything about variance, for example. This makes it difficult to have any intuition for the effect sizes. I was also surprised that there was very little detail on the inferential statistics aside from summary p-values. The amount of detail would be unacceptable in most behavioral-science outlets that I am familiar with. They may be appropriate here – I do not know.	We agree with the reviewer that presenting effect sizes is important; we regularly present r, Cohen's d, and η^2 effect size values throughout the main text. To increase clarity on the basic descriptive statistics, we have added in a new table at the start of the results section with the means and standard derivations for each of the ten composite support items that we use as dependent variables in our analysis. In response to the requirements outlined in the reporting summary for Nature journals, we have also added in additional information on F statistics for the ANOVAs and partial η^2 values.
a. Was there pre-registration of analysis plans? b. How many participants were excluded? c. Are overall demographics reported anywhere? Demographics in each country? Demographics by Global North / Global South? d. Did random assignment work appropriately across the three technology conditions? e. Again, I don't know the norms for this journal, or area more broadly, but, personally, I would like much more detail on the methods in the main text. What are the survey instruments? What are the	We appreciate the Reviewer's request for more information here, as well as the disciplinary differences in what is expected (requested) in a specialist / behavioral science journal versus one more general in orientation. While it is not possible to address all these aspects in the main text, we are more than happy to do so here. a) No pre-registration, in part because the survey was, on the whole, not focused on testing particular mechanisms but generally broader in scope. b) We are unfortunately not able to provide this information, given that

instructions? What are the technology descriptions? How did translations work? And so on. It feels “impossible” to evaluate these results without going to the Appendices. That seems problematic.
f. Will data be publicly available for verification and replication? (When and under what conditions?)

participants removed / excluded from the sample, whether from failing the quality checks, not wishing to provide consent, or just not completing the survey were not logged. This is part of the standard protocol of the professional survey firm with which we were working.

c) We would seek further guidance on this point from the editors of the journal and from the reviewer on what is requested/necessary here. Demographic data is clearly something we can provide, but with 30 countries, providing this for each individual country is clearly a substantial undertaking that we think would pay few dividends in terms of providing relevant and useful information to readers. If we are to provide additional information, we would seek to know which variables in particular, and for which populations this is deemed essential.

d) We are somewhat uncertain what the Reviewer means by “work appropriately”. In terms of number of participants in each group, yes, there are 10,084 in the “SRM” group, 10,105 in the “Ecosystems-based CDR” group, and 10,095 in the “Engineered CDR” group. Numbers were also equally balanced at a country level (i.e., there were 331-342 respondents in each of the three technology groups from each of the 30 countries – therefore, 90 sub-samples, all with somewhere between 331 and 342 respondents).

e) Thanks for raising these points – unfortunately, our hands are somewhat tied here given the limits on main text for the journal as well as

	its more generalist orientation. Overall, there is less of an emphasis on Methods being discussed in the main text. We do, of course, include the relevant material in the Appendix, which should help provide insights on such matters for any interested readers. f) In accordance with the requirements of the journal, we aim to make as much of the data publicly available as possible. Access to the raw data can be made available from the corresponding author on reasonable request, and the totality of the dataset will be made publicly available in full before the conclusion of the relevant project.
VII. Language versus Location. Translation: Was there any piloting done to compare descriptions across languages using respondents who speak both languages? Obviously this couldn't be done for every combination, but perhaps using English as a standard would provide valuable information. (I should acknowledge the ethnocentric suggestion; but it seems to be the most convenient and frequently-available comparison.) I acknowledge that this is a very difficult task, but anything that can increase confidence that we are seeing effects of location rather than survey language could be quite valuable.	We note in the Languages and Translation section in Appendix II that we followed identical procedures for translation of the base English text into all languages, including the integration of native-language experts to translate specialist keywords. Through pilot testing and soft launch with survey participants, while emphasizing opportunities for feedback, we could also identify potential issues of comprehension. We did not pilot languages at a country level – we did run preliminary analysis here (on support), finding little evidence that the language selected had a significant influence. In any case, we would note that any influence of language may not reflect (or exclusively reflect) a translation issue, but rather could serve as a stand-in for a variety of cultural factors influencing perceptions, beliefs, and reactions.
VIII. Minor questions. 1. Breaking up the paragraph that begins on 238 could be helpful for many readers. A few additional sentences that explain some	All very good suggestions, in response we have:  - Broken up the paragraph around 238 and added 1 sentence explaining the

of the connections between presuming and domestic abuse and between natural gas production and prostitution could be useful. 2. Why two panels in Figure 1? 3. Line 696 - “The findings involving age are clearly relevant for policy and developing communication strategies.” What is the policy relevance?	connection between gas and domestic abuse  - Figure 1 has 2 panels to reflect two separate studies done on Bangladesh, which is clarified in the caption to the Figure - We rewrote line 696 to be more about decisionmakers rather than policymakers specifically
SUMMARY: As I stated at the outset, these data are likely to be a major source of gravity with respect to any subsequent discussion about perceptions of geoengineering options. Such discussions are happening across fields and disciplines and well beyond academia. Not only will subsequent surveys and experiments have to ‘answer to’ (i.e., contextualize themselves with respect to) these results; but decisions about where to invest resources in technology development and governance might be influenced by these results. It is thus critically important that readers can clearly understand the contributions and limits of the current findings; and have every opportunity to calibrate their confidence in the findings appropriately.	We appreciate that, and hope that our revisions above, especially the new explicit details about methods, statistical findings, and limitations, is enough to sway the reviewer.
Reviewer #2: Thank-you for the opportunity to review COMMSENV-23-1535-T. In this manuscript, the authors describe a multi-country survey to describe the connection between demographic aspects and support for climate mitigation technologies (e.g., BECCS, DAC, biochar). Overall, the MS was well-written, but the Supplementary Materials contain superfluous information and an overly detailed description of the methods. Overall, I commend the authors for deploying a survey which encompasses both Global North and Global South countries, but do not believe this manuscript is suitable for publication in this journal. I encourage the authors to seek out a more suitable venue for publication, and perhaps adjust the scope of the MS to a briefer research note. In	Thank you for taking the time to comment on our study. It’s interesting as Reviewer 1 was criticizing us for not having enough detail in our Supplementary Materials, so we’ve sought to strike a balance in our revision. We also do respectfully maintain that the manuscript is a very good fit for this journal given that its aims and scope state that “our scope covers all areas of the geosciences, climate and environmental sciences as well as planetary sciences, including those at the interface with ecology, sustainability and environmental social sciences.” We would classify our paper as squarely within the environmental social sciences, but that is still central to the journal’s remit. We would

support of this recommendation, I provide detailed comments and suggestions below.	also maintain that because the editor sent our manuscript to peer review, when they could have desk rejected it, and then gave is a decision of Revise, rather than Reject, this is also a clear sign of fit for the journal. Lastly, we would note that both the editors and Reviewer have asked us to lengthen the manuscript, which we have done, but this does go counter (with respect) to your suggestion to write a briefer research note.
pp3 The authors note that Mahajan and colleagues warn that concerning solar radiation management techniques in particular, “there is surprisingly limited empirical support that preferences for solar geoengineering vary by gender.” The inclusion of this claim is confusing, as it appears that Mahajan et al find no difference in support by gender, and yet the authors seem to be using this claim to bolster their interest in exploring gender effects. If instead the authors are referencing this study to claim that there has been insufficient work on gender differences in support for climate mitigative technology – this should be made clearer.	We absolutely agree with this comment, we should be clearer with how we cite Mahajan et al. on p. 526. We have rewritten the sentence.
pp3 The claim “Tentative conclusions from the few studies that have examined youth perspectives on geoengineering have noted that younger people tend to prioritize climate action more strongly, but also to more strongly emphasize the need for international cooperation and governance. It is also youth that are more likely to be on social media, a platform they can use to reach millions of other individuals when they discuss climate policy or technology.” is too descriptive, and doesn’t provide a clear rationale as to why the authors are interested in the youth perspective.	Good point. We’ve added that they matter because “Youth are more vulnerable to the duration and severity of impacts of climate change than adults, given that they will generally live longer (incurring more exposure) but have physiological factors (such as smaller lungs and less developed immune systems) that make particular impacts such as air pollution or heat stress more extreme.”
pp4 The authors write “Given substantial demographic shifts currently underway around the world...”. What are these demographic shifts?	Another valid point, this half-sentence was not key to the paragraph so it’s been removed.

pp4 The authors write “Understanding how such factors enhance social diversity in climate decision-making and environmental organizations may also help speed the development of innovative technological and policy solutions.” It is not clear what this sentence means, nor does it appear to relate to the focus of the study (on demographic differences in support for climate mitigation technologies).	Again, excellent points. This sentence, and the next one, was poorly phrased. We’ve removed both, and now say this instead, with the following for support: “Understanding points of support, or opposition, across different individual perceptions can more broadly reveal patterns of incipient social acceptance or social license to operate, or patterns of anticipant opposition, both of which have high relevance for decisionmakers.” Lenton, A., Boyd, P.W., Thatcher, M. et al. Foresight must guide geoengineering research and development. Nat. Clim. Chang. 9, 342 (2019). https://doi.org/10.1038/s41558-019-0467-z Anderson, K., Buck, H.J., Fuhr, L. et al. Controversies of carbon dioxide removal. Nat Rev Earth Environ 4, 808–814 (2023). https://doi.org/10.1038/s43017-023-00493-y
pp4. Similarly for the sentence: “though understanding points of tension (and support) across different individuals can more broadly help to support more equitable and effective development and deployment processes.” Understanding points of tension within individuals does not seem to be within the scope of this study.	See above, revised to say individual preferences, rather than tensions between individuals.
pp8. While certainly there are well-established gender differences, the claims that “Women are more at risk to technology abuse and even domestic violence pertaining to the adoption of smart homes, smart meters, household energy control systems and digitalization of energy practices. Natural gas extraction and shale gas production, considered a bridge to low-emissions economies by some, also perpetuates increased rates of prostitution, sexually transmitted diseases and stillbirths, and threats to food security which more severely threaten women.” are overly specific for the scope of this study. In other	That is certainly true, we just wanted to signify why gender matters, especially insofar as it relates to the experience of violence. We’ve added more interpretation of these references to better ground and contextualize them, and also link them more strongly to our study’s focus on geoengineering.

words, I am not challenging the veracity of the forms of violence faced by women, but the specificity of the geographic and social contexts which produce this violence is not matched by the scale at which this present survey is deployed.	
pp12-13. Since this study spans the globe, it would be helpful to include data on the vulnerability of people living in poverty (vs those not living in poverty) in Global North countries as well. Further, are all comparisons illustrated in Figure 1 significant?	Good suggestion, we've added the following short paragraph with global data on poverty rates as well as for the United States (a large Global North Country): “According to the most recent data from the World Bank, approximately 9.2% of the world, or 719 million people, live in extreme poverty, or what the World Bank calculates as less than \$2.15 a day, which makes them unable to meet basic needs. Using a different estimation technique, 1.2 billion people in 111 developing or Global South countries live in multidimensional poverty, accounting for 19% of the world’s population, including 593 million children. However, poverty is a Global North problem as well, with the same data suggesting that more than 37 million people were living in poverty in the United States, of which 11.1 million were children.” We've also clarified that Figure 1 refers to a qualitative study, and thus did not present significance reporting, just descriptive statistics.
pp16, Results and Discussion. Given that there are multiple statistical comparisons made, there should be an appropriate adjustment of the significance level, i.e., Bonferroni corrections. Related, there are no details provided of the statistical analysis conducted, e.g., a U-value for the Mann-Whitney test, F and p values for the ANOVA tests, etc.	We intentionally did not report the U statistics, as these are very difficult to interpret on their own, and are exceedingly large in our analysis, due to the sample size of approximately 5,000 in each group in the gender analysis. We fully appreciate the point about multiple comparisons and have applied Bonferroni corrections for multiple comparisons to our analyses. These have led to some very minor changes (e.g., soil carbon no longer displays a significant difference for the gender Mann Whitney U test, and BECCS is no longer significant in the income Mann Whitney U test). We do, however, seek to make the point in the

	article that statistical significance is not a particularly relevant metric for analysis when dealing with a data set this large, and we always seek to draw attention to the effect sizes instead. For the ANOVAs, we have now also made clear in the main text that we have used Bonferroni corrections for multiple comparisons, when examining the results of the posthoc comparisons across the categories of the factor variables. Again, we do not think the F statistics are useful. We do report that the p-values are all lower than 0.001. The F statistics are so large that the p-values are exceedingly small. For example, in the ten ANOVAs conducted for age, the p-values had anywhere between 9 and 67 zeros preceding the first non-zero number in the decimal value. One order or magnitude does not make a difference in this situation.
pp16, Section 5.1. While the grouping of SRM technologies makes sense, I am not clear on the rationale for the other two groupings (and as described in the Supplemental material). Why was Biochar and Enhanced Weathering grouped separate from Soil Carbon Sequestration? These seem like very similar processes, akin to the SRM grouping. Further, while DAC is a fully engineered technology, BECCS involves both natural processes (plant photosynthesis) as well as combustion and carbon storage. Thus, the third grouping seems rather arbitrary.	Fair point; we now clarify that “We group carbon removal technologies based on the classifications and typologies in the literature offered by Morrow et al. (2020), Low et al. (2022), and Sovacool et al. (2022). These all distinguish nature-based solutions (afforestation, soil management, blue carbon) from engineered solutions (biochar, enhanced weathering, DAC and BECCS). While there are obvious connections between the ecosystem based and engineered carbon removal options, distinctions are made based on the degree of technical sophistication and maturity, capital intensity, and supply chains for carbon storage. Ecosystem based approaches are those that feature a more prominent role of biological, ecosystem-based sinks with a relative focus on applications on terrestrial and marine environments. Engineered approaches differ by being more technological or chemical in nature, with a relatively stronger reliance on antecedent systems of resource extraction or mining,

	carbon capture and storage as well as transportation infrastructures. Biochar and enhanced weathering represent more hybrid approaches that blur these distinctions, but we classified them as more engineered than nature-based, at least in a comparative aspect.” See: Morrow, David R., Michael S. Thompson, et al. 2020. "Principles for Thinking about Carbon Dioxide Removal in Just Climate Policy," One Earth 3, pp. 150-153 Low, Sean, Chad M. Baum, Benjamin K. Sovacool, Rethinking Net-Zero systems, spaces, and societies: “Hard” versus “soft” alternatives for nature-based and engineered carbon removal, Global Environmental Change, Volume 75, 2022, 102530 Sovacool, Benjamin K., Chad M. Baum, Sean Low, Climate protection or privilege? A whole systems justice milieu of twenty negative emissions and solar geoengineering technologies, Political Geography, Volume 97, 2022, 102702.
Further, it is not clear that a focus on 10 separate carbon mitigation technologies is warranted here. The authors do not justify why it was necessary to include all of these specific technologies in the survey, as opposed to investigating public perceptions of climate mitigation technologies more generally (pulling excess CO2 from the atmosphere, reflecting solar radiation) vs efforts that prevent the release of carbon dioxide in the first place (e.g., decarbonization). Overall, the focus on these 10 different technologies makes for an overly complicated paper. I suggest that either the Introduction be reframed to clearly justify why it is necessary to gauge support for each of the 10 technologies, or to reframe the paper as suggested.	We agree completely, great suggestion, and now explicitly done in the Introduction where we clarify: “Although much previous work has tended to look at each of these technologies by itself, or in comparison with only 2-3 other interventions, we examine all ten together as an integrated portfolio because this is how they may be synergistically deployed together as part of a future climate policy package, and because both suites of carbon removal and solar geoengineering technologies are shaping climate governance and mirrors the policymaking dilemma of choosing options with limited resources and uncertainty.”

pp16, Section 5.1.1. Blue Carbon is misspelled, and there should be a comma between Blue Carbon and DACCS	Thank you, these have been corrected.
pp17. Figure 3 is not referenced in the text, and it is not clear why there two versions of the same figure are included.	Very good catch, also corrected, Figure 3 is now cited in the text, and has only one version.
Overall, effect sizes reported in this study are small (further, it is not clear whether statistical significance has been achieved for the reported analyses, given that corrections for multiple comparisons were not mentioned). The exception may be differences in support for SRM technologies between Global North and Global South countries.	We address the reviewer’s concerns about Bonferroni corrections in our response to the comment above (and via the associated changes in the results section of the article). We would argue that whilst the results are indeed statistically significant, even with the corrections for multiple comparisons, statistical significance means quite little when dealing with sample sizes this large. We had p-values with fifty zeros following the decimal point, but still with only moderate effect sizes. Indeed, we agree with the reviewer that the effect sizes are quite small in several instances (e.g., for gender and income). Nevertheless, we feel these are valuable findings. For example, we specifically draw attention to the small effect size for gender when we write in the Conclusions section: For gender, perhaps the most intriguing finding is how little effect this demographic characteristic had overall. All the effects from gender, when examined on its own, were small... We go on to explain that for gender, previous authors have anticipated, predicted, and revealed much larger effects; we are drawing attention to a different set of effects. For income, whilst the effect sizes are small, we think the findings matter because the most common expectation in the foregoing literature on this topic is that the relationship would exist in the opposite direction. Therefore, it is not the magnitude of the effect, but the direction of the relationship that we found that matters here.

In both the Results and Discussion, and in the Conclusion, the authors focus more on describing the results rather than on highlighting how these results can inform our approach to addressing climate change as a global community, how they inform the communication best practices around carbon mitigation technologies, how they can provide insights for policy rollout, etc., etc. This should be the most important part of the Discussion/Conclusion for this paper.	We thank the reviewer for pointing out this oversight, and suggesting the opportunity to clarify the policy and communication relevance of our research. Indeed, we agree that there decidedly is practical value to our results. We have added additional text in multiple areas of the Conclusion section in the article, immediately following our summary of the core findings on gender, age, and Global South vs Global North. These three added blocks of text point to the communication and policy implications of the core findings.
Additionally, if there are indeed statistically significant demographic and intersectional identity effects, what might explain these particular results? In light of the detailed literature review provided in the Introduction on these demographic aspects, there are few new insights provided from this globally deployed survey (as currently described).	We note in the Conclusion section that few statistically significant intersectional findings emerged. We do discuss the intersectional findings in the Conclusion (e.g., males and those in poverty show stronger effects of age for the nature-based CDR approaches). Nevertheless, we seek to make the point in the article that it is the main effects that are most prominent. In our response to a comment above, we note that one key finding that diverges from previous research is the lack of effect of gender on support for CDR approaches, which could increase viability of and distributive fairness of policies to promote CDR interventions. In that same comment above, we also note that poverty and living in the Global South both have the opposite effect of what has been suggested by foregoing research literature in this area. This is certainly a new insight. This matters for actors (government or otherwise) considering implementation of CDR projects in low income and/or Global South contexts.
Finally, I am curious why the authors did not include values and/or political orientation (not specific parties, as that would be inappropriate – but whether an individual is more liberal or conservative), given that these are significant explanatory variables in belief in/concern about climate change,	Political orientation can indeed be an important predictor of climate change views in some national contexts, although multiple studies have shown that this relationship is strongest in the US, then still manifest in Western Europe (although weaker), and quite weak or non-existent in much of the

support for climate change policies, etc., I am surprised that they weren't included in this study.

rest of the world. One concern with this variable is reliability of how the wording is interpreted across the countries in a truly international study. We would not be able to use the language of 'liberal' and 'conservative', even if the study were only confined to Europe, as these words have very different meanings outside of the US context. We have used 'politically left-leaning' and 'politically right-leaning' in some previous studies to try to capture this construct, but it is still quite difficult to make comparisons across countries. For example, these terms can still be difficult to interpret and understand materially in countries ruled by complex coalition governments in parliamentary systems.

We do have a fairly broad scope measure of political orientation in our data set that we could include in the analysis if the reviewer and editors feel strongly that this additional analysis should be provided, but we think it would be better left out for a few reasons:

- we focus on socio-demographic, rather than attitude/value variables in the current article;
- political orientation is difficult to compare across nations (perhaps in a similar way to how poverty/income is hard to compare, but even to a greater extent); and
- we are concerned that cultural, political structure, and perhaps even language/translation differences between countries make the reliability of this measure uncertain.

Figure 3 (reconfigured): Support for climate interventions by gender, with composite support variable subdivided into its three components

Figure 5 (reconfigured): Support for climate interventions by income, with composite support variable subdivided into its three components

Editor and reviewer comments	Response from the authors
Summary of revisions	We appreciate the opportunity to revise again and improve the paper and have made multiple substantive revisions in response to the referee comments. We believe the revised manuscript fully meets the first criterion about insights into the demographic dimensions of public preferences for the 10 climate intervention technologies examined. To clarify, we would not classify our study’s research design as experimental or quasi-experimental. It has no control group, nor any sort of choice experiment. (We considered this, but ruled it out due to lack of our own expertise and lack of resources). Instead, we would classify our study as falling into the “Surveys and quantitative data collection” category (according to https://www.sciencedirect.com/science/article/pii/S2214629618307230), which is distinct and has different codes of practice, but is very well established in the social sciences literature. That said, we have nevertheless better justified our variables and selection of technologies in our revision, and provided more details about significance tests run.
Reviewer #1: I really admire this research effort. I appreciate the improvements made to the revision on many dimensions. I believe we can learn a lot from these data. However, in my opinion, the current writeup remains problematic. 1. It is not at all clear if these results generalize beyond these specific descriptions of the different technologies, and, in my view, this fundamental issue is not addressed clearly in the manuscript. The core question here is how different people (specifically different demographic groups of people) evaluate a stimulus. But the stimulus here is	We appreciate the reviewer’s comment and understand their hesitancy in relation to some of the claims in the article, due to the results depending on the wording of the information provision. Indeed, in any survey, interpretation of the results will always be based on the wording of the questions. In the case of our survey, this issue comes more to the fore than in some other surveys, due to the heavy relevance of information provision (i.e., introduction of new information, as opposed to just having the wording choices relating to the text of the questions themselves).

one authorship team’s descriptions of the technologies, not the technologies themselves. There is no empirical reason to believe that the stimulus is representative of the class of stimuli which it is supposed to stand for. I suspect that I (or anyone) could write different (reasonable) descriptions of the same technologies and find very different results. Of course, then reasonable people could and should debate which descriptions are more reasonable / appropriate / representative etc. But any reader should have serious questions about how important the specific details of the descriptions are, and my concern is that the writeup is not clear enough about this limitation of the work. I fear that many readers (scientists, media, and public) will overinterpret based on the current discussion. This concern is exacerbated by frequent declarations about the policy relevance of these results. (For example, if we include even minimal information about risks and uncertainties, we might see very different patterns and levels of support.) I believe strongly that this limitation of the research design should be front and center in every major section of the paper (abstract, introduction, methods, discussion). I am not suggesting that the researchers can or should go back and re-run with multiple descriptions – that is impractical at this stage. It was a reasonable decision for them to make for all of the reasons they elaborate in the letter. But, in my opinion, it is essential that it be discussed openly and clearly as a methodological choice that limits interpretation. Readers might be somewhat reluctant to accept that there is “substantial policy relevance” from one result based on one description. The ambiguous sentence that, “This complicates [interpretation] in some ways...” is not adequate in my view.	We agree with the reviewer that we were careful about our selection of the information we provided, and that at the same time, different information certainly could have been offered. An alternative approach to the design of this study could have involved much more extensive information provision on only one of the carbon removal approaches. That could have ensured more nuance and depth in the information provided, but would have not allowed for the comparative aspect that interested us in our research. We fundamentally agree with the reviewer that irrespective of what wording was chosen, reactions by survey respondents to the text could always be different if different text was used. As such, we have included substantially more qualifications about this throughout the manuscript (in the abstract, introduction, research design, results, and conclusion sections – see manuscript with track changes highlighted).
2. I remain concerned that the conclusions about specific technologies may not be	We now speak to this point explicitly in the research design section of the manuscript.

warranted given the nested research design. Individual technologies were completed nested within groups of technologies. (Again, this was a reasonable decision based on response fatigue, etc.) But treating them as independent in analysis and conclusions therefore seems inappropriate. Maybe I am off base here, but – given the likely influence of this important work – I highly recommend that an additional reviewer who specializes in data analysis be consulted in the process before publication (if one has not been already).	We provide reference to methodological literature that supports the comparative analysis we undertake, due to the combination of probability sampling and random assignment across the experimental conditions of the information provision. We also now return to this point in the results and conclusion sections.
3. Ignoring the nesting problem, it is not clear to me that the results merit the statement that there is “support on average” for the technologies. If I am reading correctly, few if any of these means are significantly different from the scale midpoint. Was this tested? For many of the technologies, it seems just as accurate to assert that the means indicate “neither rejecting nor supporting on average.”	We can understand the reviewer’s desire for caution here. The scale ranged from 1 to 5, with 3 as the scale mid-point. For all ten climate interventions, the lowest mean value was 3.33. Although in total size this might not seem much larger than 3.00, due to the very large sample size in our survey, the 95% confidence interval for this mean estimate was 3.30 – 3.35. For each of the ten mean estimates, the entire 95% confidence interval was smaller than 0.05. By this measure, each of the means is substantially and unequivocally above the scale mid-point. We have now added this 95% confidence interval data to Table 1. The reviewer, however, is correct to point out that the definition of 3.00 on the scale is ‘neither reject nor support’, whilst it is 4.00 that is defined as ‘somewhat support’; therefore, only the three nature-based CDR approaches are clearly in the realm of support. We now have changed our language (in the text just prior to Table 1) to indicate the nuance that the other seven approaches have means positioned between ‘neither reject nor support’ and ‘somewhat support’.
4. I am confused by the paragraph (p. 16 on my view) discussing “priming.” How can it be that the materials avoid priming	We fully agree that these sentences did not make sense together. We have reworded the second sentence to read:

participants with risks versus benefits and also that the descriptions offer a balanced presentation of risks and benefits? I recognize and appreciate that this was added in an effort to address one of my questions, but I have to admit that I am confused by the idea here. In any event, I highly recommend – for this reason and comment (1) above – that the descriptions be included in the main text instead of requiring readers to go to SOM. It seems that they could fit in a table without taking up too much journal space.	‘Even though our technology descriptions (shown in Supplemental Materials) offered a balanced presentation of the extent to which they might present viable climate solutions, due to uncertainty or lack of knowledge respondents may nevertheless err on the side of undervaluing potential benefits.’ Indeed, we did not provide information on risks; we provided functional information about how the interventions worked, including information (both promising and concerning) on their prospective efficacy, or limitations thereof. In terms of including the full descriptions in the main text of the article. We would be very supportive of this, and agree they could be helpful to the readers. They are currently in the appendix because all ten descriptions together comprise 1,200 words of text and ten images. Before moving this to the main text of the article, we defer to the editors as to whether this is something they would accept – and also, whether this might affect the flow of the article.
I expect this work to have substantial influence. I hope that the published version will be written up in the most transparent and balanced form possible to help maximize learning and minimize misinterpretation. I believe that the authors share this goal and I hope these comments help push them further in this direction.	Thank you for your supportive comments and genuine efforts to help us improve the manuscript. We agree that both the tone and clarity of the manuscript is now further enhanced.
Reviewer #3: Thank-you for the opportunity to review the revised MS COMMSENV-23-153A. As before, I commend the authors for deploying a survey which encompasses both Global North and Global South countries, but still do not believe this manuscript is suitable for publication in Nature Communications: Earth & Environment. I encourage the authors to seek out a more	We continue to believe that the manuscript is well suited to Communications: Earth & Environment. We do not believe data from a survey of over 30,000 respondents, stratified across 30 countries and conducted in 19 languages, and analysed for intersectionality across four sociodemographic variables is appropriate or feasible to cover within a ‘research note’.

suitable venue for publication, and/or adjust the scope of the MS to a briefer and more focused research note. In support of this recommendation, I reiterate my main concerns below.	
The authors present a long literature review detailing vulnerabilities to, and concern with, climate change across demographic and socioeconomic categories (and resulting intersectional identities). However, there remains scant theoretical justification for why one may expect difference in support for 10 different forms of geoengineering and carbon capture technologies across these different sociodemographic groupings. Overall, the introduction provides insufficient justification for the study design, analyses, and interpretation of results. The introduction should be rewritten to provide a clear, logical, and succinct justification for why we should expect demographic and socioeconomic differences in support for specific geoengineering and carbon capture technologies. For example, I believe that there are interesting reasons to focus on differences between youth and adults in support for geoengineering and carbon removal technologies. However, the justification for the focus on youth in this study is still insufficient and overly general. To say that “Youth are more vulnerable to the duration and severity of impacts of climate change than adults, given that they will generally live longer (incurring more exposure) but also presently have physiological factors (such as smaller lungs and less developed immune systems) that make particular impacts such as air pollution or heat stress more extreme.” Certainly justifies a focus on risk perceptions and, perhaps, support for actions to mitigate climate, but not on why youth might differ in their support for specific types of geoengineering and carbon capture technologies. Similarly, while the authors	Our research was motivated by an appreciation of the importance of intersectionality broadly, and then the desire to see how this was manifest (in an exploratory sense) in an extremely large and robust data sample. The literature review shows how there is equivocation in the literature over these intersectional relationships, and our goal was to offer additional clarity to that conversation, as opposed to answering a specific hypothesis. Indeed, the extant research literature itself does not offer clarity on the roles that each of these sociodemographic variables does, or theoretically should, play in shaping views towards climate interventions. This led us to explore such relationships in our large and diverse survey sample, to offer further evidence on these relationships. The analysis was exploratory, not hypothesis-driven.

mention gendered differences in support for, perceptions of, and use of technology, I don't see a clear connection to the types and groupings of the technologies used in this study.	
Related, the MS still lacks sufficient justification as to why it is necessary to conduct a survey examining 10 specific types of geoengineering and carbon capture technologies on a global scale. Further, these technologies were organized into three groups SRM (stratospheric aerosol injection, marine cloud brightening, space-based geoengineering); ecosystem-based CDR (afforestation and reforestation, soil carbon sequestration, marine biomass and blue carbon); engineered CDR (direct air capture with carbon storage (DACCS), bioenergy with carbon capture and storage (BECCS), enhanced weathering, biochar). I mentioned in my initial review that the groupings were inadequately justified, and I reiterate that concern again. The authors point to three studies to justify their groupings (Morrow et al. (2020), Low et al. (2022), and Sovacool et al. (2022)). However, these studies do not appear to match the carbon removal technology groupings used in the present study; these published studies used, for example, "Nature-based carbon dioxide removal (e. g., afforestation and reforestation, biochar, soil sequestration, blue carbon and seagrass, ecosystem restoration)" and Engineered carbon dioxide removal (e. g., carbon capture and utilization and storage, bioenergy with carbon capture and storage, ocean iron fertilization, enhanced weathering, direct air capture) or they used "Nature-based, Hybrid and Engineered carbon removal". Further, many of these technologies are unproven and have had limited application and discussion outside of policy and academic circles. I thus have concerns with what participants (particularly those with limited literacy and	In response to the comments above from reviewer #1, we have added additional language throughout the manuscript (i.e., in the abstract, introduction, research design, results, and conclusion sections) to be explicit that the survey respondents are likely mostly unaware of the climate interventions they are assessing, and that their responses are necessarily dependent on the information provided to them via the survey. We do not contest that one could have grouped these technologies / interventions differently, though we would note that we had substantive reasons for grouping them as we did. Our argument is not that our grouping them into the sets of three, three, and four approaches is the only correct way to group them, but merely that the groupings are one sensible approach to breaking up the ten interventions so that the amount of text and questions presented to each respondent would be manageable and not lead to satisficing and survey fatigue. We would also note that, having discussed and tested out alternate grouping approaches (e.g., one SRM, one ecosystem-based CDR, and one engineered CDR) in each category, we determined that such groupings were more likely to be confusing, by presenting participants with technology options varying in terms of underlying approach, not to mention "maximizing" the perceived differences between the options presented.

education) might be responding to in these scenarios given their low to no exposure outside of this survey setting (i.e., survey response effects).	
I remain unsatisfied with the reporting of statistical analysis. The authors have provided insufficient justification for the tests used, and in the reporting of results. Specifically, why were both parametric and nonparametric tests used? Given the lack of justification for a focus on intersectional identity effects and support for geoengineering and carbon capture technologies, why were three-way ANOVAs conducted (as opposed so one-way, or two-way)? How was education level (and other confounding variables) controlled for in these tests? Overall, there remains insufficient detailing of the analyses used beyond the reporting of effect sizes.	In terms of the parametric versus non-parametric tests, we conducted both the Kruskal Wallis H tests and the ANOVAs because, although the non-parametric tests may be more statistically appropriate, we believe that the results of the ANOVA are much more accessible and easier to interpret for an interdisciplinary readership, who might not have substantial background in statistics. We focus on the more accessible data in the main text of the article and then run the non-parametric tests to ensure that the parametric results are still accurate representations of the data – notably, if performing one or the other led to a change in findings of significance or large differences in coefficient estimates. Having found this not to be the case, we were confident in retaining results from the more accessible modeling approach. In our response to a previous comment by reviewer #3, we mention that our interest in intersectionality broadly, and the lack of research on how intersectionality shapes responses to climate change response strategies, informed our research design. This was not driven by a specific hypothesis. We agree with the reviewer that examining effects of education (either independently or in conjunction with the other sociodemographic variables) could be revealing. We feel, however, that based on the amount of analysis already present in the current article, this additional analysis would be best reserved for future research employing this data set.
Finally, while a focus on poverty is interesting, I am unclear as to why the	We felt it important to offer some analysis of the potential effects of poverty on reactions

authors used a poverty threshold established for higher income countries. Further, given that most countries in the study had zero participants below the poverty level, do country level effects not confound the effect of poverty? Overall, only 3.3% of survey respondents could be categorized as living in poverty, and these came from 8 countries (as six of these eight countries were from the Global South, that is also a confounding factor). Further, the authors explanation for how the poverty threshold was arrived at is confusing as written.

to the climate interventions, due to the literature that equivocates on this topic. Nevertheless, we fully accept that this analysis is necessarily a first attempt, and that any approach that seeks to compare poverty across vastly different countries will need to gloss over much of the nuance that is needed when examining how poverty operates within each national context.

We now include the following text just before Table 2, to better qualify the analysis that we then present:

‘We acknowledge that what poverty means across the thirty different countries varies widely, and that there is likely notable additional variability within individual countries. We selected a relatively high poverty definition to capture as many relevant respondents as possible in this definition. Noting that all of our respondents in poverty come from only eight of the thirty countries, the country of residence clearly has an effect on whether someone meets our definition of being in poverty or not (see notes for Table 2). This poverty analysis is admittedly imperfect, but is a first attempt at offering empirical evidence that begins to shed light on the relationship between poverty and reactions to climate interventions – to help inform future research directions.’

Additionally, we offer additional examples in the notes to Table 2, to better explain the approach we took to calculating whether a respondent met the poverty level threshold or not.

Reviewer comments	Response from the authors
Reviewer #1: This revision has again improved on multiple dimensions. The handling of “descriptions” of technology at appropriate sections in the manuscript is clear and straightforward and I think does a great service to helping readers interpret the findings appropriately. Improved precision in interpreting the mean values is similarly helpful.	Thank you, we are pleased the manuscript continues to improve and that you appreciate the effort put into earlier revisions.
2. I do not yet understand how the nesting issue has been addressed (i.e., implied cross-group comparisons based on fully confounded group presentation). Imagine we want to know, How healthy is Coca Cola? Imagine Survey A: How healthy or unhealthy are each of the following beverages: Coffee, Tea, Matcha, Coca Cola. Imagine Survey B: How healthy or unhealthy are each of the following beverages: Frappuccino, Milkshake, Coca Cola. It is possible that Coca Cola comes out looking worse in the first group than the second group based on its relative standing in the group. Now imagine you want to know how people rank the healthiness of the following beverages Coffee, Tea, Matcha, Frappuccino, Milkshake, and Coca Cola. An experiment that seeks to rank Coca Cola based on the combined results of the following two groups {Coffee, Tea, Matcha}, {Frappuccino, Milkshake, Coca Cola} MIGHT produce a different ranking than an experiment that seeks to do so based on the groups {Coffee, Tea, Matcha, Coca Cola}, {Frappuccino, Milkshake}. Neither the analyses nor the interpretations acknowledge this potential source of bias. If Van den Brakel (2019) puts this issue to rest, I do not understand how and explanation	We wish to thank the reviewer for the time they invested in lucidly explaining this issue. We have added the following text, at the end of the second paragraph of the results section, where we previously discussed the grouping of the technologies that survey respondents saw: The respondents rated each technology on a scale of 1-5 (strictly reject to strongly support); they were not asked to rank order their preferences for technologies. Whilst it remains possible that information from one technology could have influenced responses to the other technologies, or that questions about one technology led to an ‘anchoring and adjustment heuristic’, we believe the rating scale approach used always for reasonable comparison across all ten technological approaches. The main point we seek to make here, in contrast to the example that the reviewer provides in their comments about beverages (which indeed made some of us thirsty), is that a rating scale provides a different level and type of information to that provided by a ranking approach in survey questions. We do not contest that one could have grouped these technologies / interventions differently, though we would note that we had substantive reasons for grouping them as we did. Our argument is not that our grouping them into the sets of three, three,

might be appropriate. The discussions on pp.18-19 appropriately address potential “extraneous factors” that reside within the participants (or different groups of participants), but not extraneous factors attributable to stimulus grouping. Steps like breaking Table 1 into separate tables or panels defined by the four groups would, in my opinion, help to discourage / decrease comparisons potentially inappropriate cross-group comparisons. Explicitly acknowledging this issue would be even more helpful to readers.	and four approaches is the only correct way to group them, but merely that the groupings are one sensible approach to breaking up the ten interventions so that the amount of text and questions presented to each respondent would be manageable and not lead to satisficing and survey fatigue. We would also note that, having discussed and tested out alternate grouping approaches (e.g., one SRM, one ecosystem-based CDR, and one engineered CDR) in each category, we determined that such groupings were more likely to be confusing, by presenting participants with technology options varying in terms of underlying approach, not to mention “maximizing” the perceived differences between the options presented. As far as breaking apart Table 1, we believe this would not be holistically helpful for a few reasons. First are the arguments and explanations above. Second, this would necessitate that any such visual presentation of results for different groups would demand different tables unless the group was asked about all options. Yet, there are papers in the literature where ten technologies are considered, and compared, with each participant randomly assigned only one. One could argue that then there are no issues with "grouping", which is true, but our response above about rating scales vs ranking scales speaks somewhat to this concern. We also checked on ordering effects (see response to R1 in first round of revisions), for which we found limited evidence. It bears emphasizing that we randomized the order in which technologies were presented to any given participant to avoid this as a source of bias.
3. I do not understand the following argument: “Nevertheless, as is necessarily the case when providing information, had different information been provided, respondent	We can appreciate the confusion here, as we did not previously explicitly state that these differences related to age. The second sentence was wordy and ambiguous. We

evaluations of the interventions could have differed. We however note that, having ruled out any (unintended) significant differences among the groups randomly assigned to the technology categories, we cannot identify material reasons for why such differences should have occurred.” How does the second sentence bear on the first?	have removed it and replaced it with the following: We have no reason, however, a priori or from the data itself, to suggest why the information we provided on the technologies would lead to any of these age-related effects.
4. I do not know the journal’s policy, but I wonder if a final version of this article should address how it differs from and builds on the now-published work by Baum and colleagues (2024) in Nature Communications.	We appreciate the reviewer’s feedback and the opportunity to clarify the differences and similarities between our work and the study by Baum et al. (2024). While the point of departure (i.e., the macroscopic raw data set) for our study is indeed the same as in Baum et al. (2024), our focus diverges significantly. Specifically, we aim to understand why there is a pronounced difference in support for climate intervention technologies between intersectional demographic characteristics. To achieve this, our analysis utilizes individual-level data rather than country aggregates, as was done in Baum et al. (2024) – the previous study also focused exclusively on Global North versus Global South regions, rather than considering differences by demographics – gender, age, and socioeconomic status. We have added the following text to the end of the penultimate paragraph in the methods section: Results of the full dataset, arising from the survey across the 30 countries, have been reported elsewhere (Baum et al. 2024); however, that prior analysis analysed data only aggregated at the country level, whereas the focus in this article is heavily on the relationship between multiple individual-level demographic variables and support for the climate intervention technologies.
5. There seems to continue to be conflicting information about whether this work should be viewed as an experiment or not. The	Thank you for pointing out this lack of clarity. Indeed, we feel that, especially in an interdisciplinary journal, our approach

response letter takes a strong stance that it is not an experiment, but then refers to it as an experiment a few lines down. The manuscript talks about embedded experiments. I don't have a strong stance, but it is a little confusing.	should not be called experimental (to not create confusion when compared with experiments, for example, in cognitive psychology). Therefore, we believe that varying 'information conditions' is a more appropriate and less confusing way to discuss our approach. We have updated any instance in which the word 'experiment' was used in the article.
6. I am confused by this sentence: "For income levels and poverty, the most basic observation is also one of the most striking – that those in poverty were nearly universally more supportive of climate interventions of various types, compared to people not in poverty" What does "nearly universally" mean here? The universe of technologies? Of participants? It seems like it could easily be misinterpreted either way. If it is referring to distributions of opinion among those in poverty versus those not in poverty, is it really near universal? Do the distributions not overlap substantially?	We have revised this sentence to increase clarity and avoid any potential for misinterpretation: For income levels and poverty, the most basic observation is also one of the most striking – that on aggregate those in poverty were more supportive of climate interventions, compared to people not in poverty (observed for four of the ten interventions, with the other six interventions showing no relationship in either direction).
Reviewer #4: This article contributes to a small but growing literature on perceptions of carbon dioxide removal and solar radiation management outside of a few well-studied countries in the Global North. The scope of the work is impressive, and the methods rigorous. I find the authors' responses to reviewer comments adequate, barring the question of the poverty level cut-off—which, as I see it, is central to their analysis, and thus warrants revision or further justification.	Thank you for your comments on our revisions; we discuss the poverty level cut-off below.
While I appreciate the authors' acknowledgement (in the rebuttal and in the manuscript) that poverty must necessarily be flattened to be compared in an analysis like this, I am uncomfortable with the extent to which the authors flatten it—especially	We agree with the reviewer that it makes sense to signpost further the limitations with the poverty measure, and to increase emphasis on the age and gender findings whilst reducing focus in the conclusions

given that the authors present the role of poverty in perceptions of climate interventions as one of their primary contributions. Their current measure is a poverty threshold (even if a conservative one) for Global North countries. As reviewer #3 notes, there are two problems with this. The first is that this threshold ends up being artificially high for developing and lower-income countries, and the second is that poverty in Global North countries is poorly represented given the higher [universal] threshold (and, presumably, sampling). I am not convinced that the findings in this paper demonstrate something about the role poverty plays in perceptions of climate interventions that is different from the finding already reported in Baum et al. (2024) in Nature Communications that publics in the Global South show significantly greater support for climate intervention strategies than those in the Global North. While the findings on gender and youth are interesting, if these are to become the central tenets of the paper, the authors should better explain the importance/implications of these findings to make up for the loss of the poverty metric. It is also unclear to me why education wasn't included in the battery of covariates, as that (along with political orientation and, depending on the national context, race and/or ethnicity) is very often used in analyses like these, as it often shapes perceptions of climate issues.

section on the poverty findings. We have now made these adjustments.

We believe the key difference in the poverty findings vs the Global North / Global South findings is that poverty is an individual level measure, whilst Global North vs Global South is a group (country) level measure. The pattern is certainly the same, but the evidence for the relationship existing is strengthened by the data coming from individual level and group level variables. We have also clarified this point in the conclusions section.

We very much accept that our measure of poverty is extremely crude and that more research on this relationship needs to be undertaken. Unfortunately, whilst our sample was very intentionally constructed on several metrics, it was not specifically designed to account for or represent nations on poverty distribution in each country. Future research where the primary focus is poverty could undertake a more fine-grained approach to representing and defining poverty in its sample construction.

For good measure, we have also clarified, explicitly and in detail, how our study differs from that of Baum et al. (2024) – even though it uses the same macroscopic dataset, it has entirely different lines of argument, and indeed authors. Specifically, we aim to understand why there is a pronounced difference in support for climate intervention technologies between intersectional demographic characteristics. To achieve this, our analysis utilizes individual-level data rather than country aggregates, as was done in Baum et al. (2024) – the previous study also focused exclusively on Global North versus Global South regions, rather than considering differences by demographics – gender, age, and socioeconomic status.

	We have added the following text to the end of the penultimate paragraph in the methods section: Results of the full dataset, arising from the survey across the 30 countries, have been reported elsewhere (Baum et al. 2024); however, that prior analysis analysed data only aggregated at the country level, whereas the focus in this article is heavily on the relationship between multiple individual-level demographic variables and support for the climate intervention technologies. Education and political orientation were not included to retain the core focus of the manuscript, and also because when we did include those aspects the results of the paper did not change significantly, i.e. those factors were not seen to have any strong correlational significance with our results.
In conclusion, while I would like to see this article published, I am not convinced by the poverty analysis and fear that it may be a reiteration of an analysis that has already been published with this dataset.	We appreciate the reviewer's attention to the more coarsely-grained and tentative nature of the findings in relation to poverty, and have made the aforementioned changes to the manuscript in response. Because the poverty data is individual-level, however, compared to the group-level Global North / South data, we do feel that the poverty analysis adds, at least incrementally, to the literature in this area of scholarship.